# Comparative neuroimaging of sex differences in human and mouse brain anatomy

Elisa Guma[1]*, Antoine Beauchamp[2,3,4], Siyuan Liu[1], Elizabeth Levitis[1], Jacob Ellegood[2,3], Linh Pham[1,5], Rogier B Mars[5,6], Armin Raznahan[1]*†, Jason P Lerch[2,3,4,5]*†

[1]Section on Developmental Neurogenomics, Human Genetics Branch, National Institute of Mental Health, Bethesda, United States; [2]Mouse Imaging Centre, Toronto, Canada; [3]The Hospital for Sick Children, Toronto, Canada; [4]Department of Medical Biophysics, University of Toronto, Toronto, Canada; [5]Wellcome Centre for Integrative Neuroimaging, Nuffield Department of Clinical 15 Neurosciences, University of Oxford, Oxford, United Kingdom; [6]Donders Institute for Brain, Cognition and Behaviour, Radboud University Nijmegen, Nijmegen, Netherlands

**\*For correspondence:**
elisa.guma@mail.mcgill.ca (EG);
elisa.guma@mail.mcgill.ca (EG);
raznahana@mail.nih.gov (AR);
jason.lerch@ndcn.ox.ac.uk (JPL)

†These authors contributed equally to this work

**Competing interest:** The authors declare that no competing interests exist.

**Abstract** In vivo neuroimaging studies have established several reproducible volumetric sex differences in the human brain, but the causes of such differences are hard to parse. While mouse models are useful for understanding the cellular and mechanistic bases of sex-specific brain development, there have been no attempts to formally compare human and mouse neuroanatomical sex differences to ascertain how well they translate. Addressing this question would shed critical light on the use of the mouse as a translational model for sex differences in the human brain and provide insights into the degree to which sex differences in brain volume are conserved across mammals. Here, we use structural magnetic resonance imaging to conduct the first comparative neuroimaging study of sex-specific neuroanatomy of the human and mouse brain. In line with previous findings, we observe that in humans, males have significantly larger and more variable total brain volume; these sex differences are not mirrored in mice. After controlling for total brain volume, we observe modest cross-species congruence in the volumetric effect size of sex across 60 homologous regions ($r=0.30$). This cross-species congruence is greater in the cortex ($r=0.33$) than non-cortex ($r=0.16$). By incorporating regional measures of gene expression in both species, we reveal that cortical regions with greater cross-species congruence in volumetric sex differences also show greater cross-species congruence in the expression profile of 2835 homologous genes. This phenomenon differentiates primary sensory regions with high congruence of sex effects and gene expression from limbic cortices where congruence in both these features was weaker between species. These findings help identify aspects of sex-biased brain anatomy present in mice that are retained, lost, or inverted in humans. More broadly, our work provides an empirical basis for targeting mechanistic studies of sex-specific brain development in mice to brain regions that best echo sex-specific brain development in humans.

## eLife assessment

In this **important** study, Guma and colleagues describe the use of structural neuroimaging to assess the cross-species convergence of sex differences in global and regional brain volumes in humans and mice. The goal of the work is to inform to what extent mouse studies of these aforementioned sex differences have relevance to humans. The authors suggest which aspects of brain anatomy (as

measured by volume) are conserved or not, across species, which has theoretical and practical implications beyond a single sub-field. The evidence to support the findings is **solid**, it uses methods and data analysis that are appropriate and validated.

## Introduction

Humans show numerous sex differences in the prevalence, age of onset, and presentation of brain-related conditions (*Bao and Swaab, 2010*). Early onset neurodevelopmental conditions, such as autism spectrum disorder, attention-deficit/hyperactivity disorder, Tourette syndrome, and language impairments tend to disproportionately affect males. Adolescent and adult-onset conditions such as depression, anxiety, eating disorders, and Alzheimer's disease tend to disproportionately affect females (*Bölte et al., 2023*). There is also evidence from multiple large-scale studies for sex differences in certain cognitive and behavioral traits such as in language and face processing (*Herlitz and Lovén, 2013*; *Olderbak et al., 2019*), spatial rotation (*Lippa et al., 2010*), and aggression (*Archer, 2004*). These observations may reflect sex differences in brain organization arising from a complex mix of genetic and environmental influences. To date, the largest studies testing for sex differences in human brain organization have focused on anatomical measures extracted from in vivo structural magnetic resonance images (sMRI). While there is considerable heterogeneity in the findings of this literature (*Eliot et al., 2021*), owing potentially to variation in the methods used (*Zhou et al., 2022*), there are several large-scale studies that recover highly reproducible sex differences in regional human brain volume above and beyond sex differences in total brain size (*DeCasien et al., 2022*). These include larger limbic, and temporal regional volumes in males, and larger cingulate and prefrontal regional volumes in females (*Liu et al., 2020*; *Lotze et al., 2019*; *Ruigrok et al., 2014*; *Williams et al., 2021*).

Gaining a deeper understanding of the causes and consequences of sex differences in the human brain is challenging due to its relative inaccessibility, inability to perform invasive experiments, and potential environmental confounds. Significant advances in our understanding of sex differences in regional volume of the mammalian brain have come from rodent studies. This literature provides an important context for thinking about volumetric sex differences in the harder to study human brain. Highly robust sex differences in the regional volume of the rodent brain have been historically identified using classical histology (*Gorski et al., 1978*; *Hines et al., 1992*; *Kim et al., 2017*). These differences have also been recovered using sMRI methods, analogous to those used to study regional sex differences in human brain volume (*Qiu et al., 2018*; *Spring et al., 2007*). These histologically- and sMRI-resolvable sex differences in regional volume of the rodent brain include a larger volume of the bed nucleus of the stria terminalis (BNST), the medial amygdala (MeA), and the medial preoptic nucleus (MPON) in males. In addition to these canonical sex differences, sMRI has uncovered several other sex differences in regional brain volume (*Qiu et al., 2018*; *Spring et al., 2007*; *Wilson et al., 2022*) including larger anterior cingulate cortex, hippocampus, and olfactory bulb volume in males and larger cerebellum, midbrain, caudoputamen, thalamus, and cortex volume in females. Modern tools for brain-wide histology in mice have established that foci of volumetric sex differences from sMRI are also salient foci of sex differences in cellular composition (*Kim et al., 2017*). Furthermore, they are concentrated within circuits subserving sex-specific reproductive and social behaviors in mice. Third, beyond allowing a paired description of sex differences in gross volume (using sMRI) and cellular composition (using histology), mice also enable mechanistic dissection of regional sex differences through genetic and environmental manipulations. For example, the four core genotype (FCG) model, in which the complement of sex chromosomes (XX vs. XY) is made independent of gonadal sex (testes vs. ovaries), has allowed researchers to appreciate the differential effects of gonadal sex (presence of testes or ovaries independent of chromosome complement) from sex chromosome complement (XX vs. XY mice of either gonadal sex) (*Arnold and Chen, 2009*; *McCarthy and Arnold, 2011*).

The above considerations drive a pressing need for systematic comparison of volumetric sex differences between the human and mouse brain. Such a comparison would provide two critical outputs. First, it would advance the understanding of brain evolution by formally testing for the conservation of sex-specific brain organization between two distantly related mammals. Second, any homologous brain regions that show congruent volumetric sex differences in humans and mice would represent high-priority targets for translational research - leveraging research opportunities in mice to scaffold studies on the causes and consequences of sex differences in the human brain.

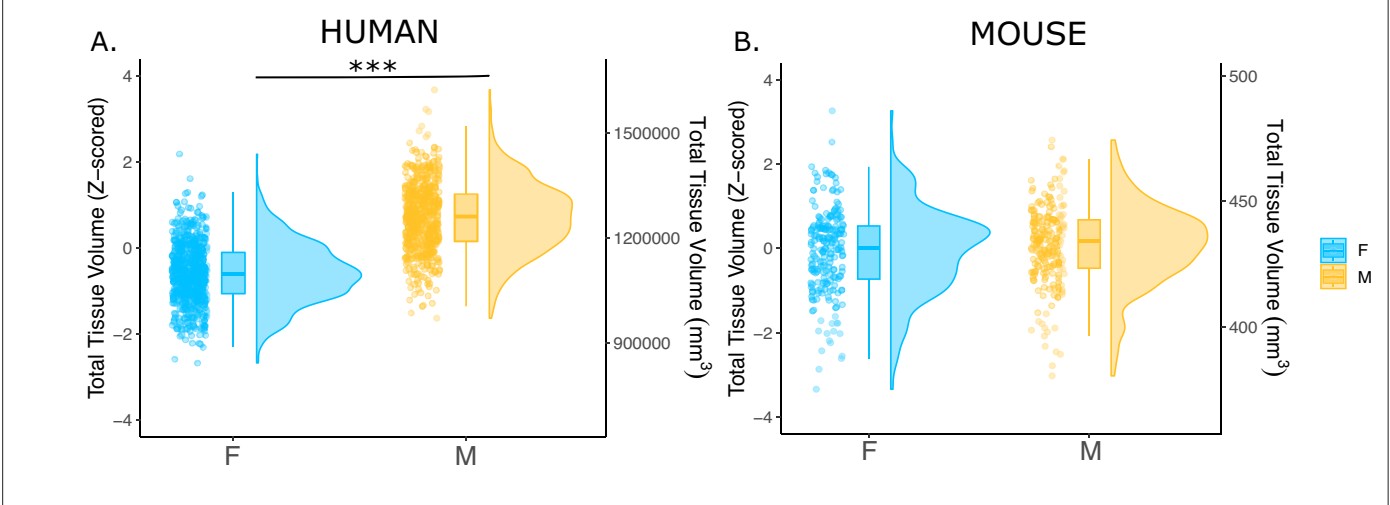

**Figure 1.** Effects of sex on total tissue volume (TTV) in humans and mice. Distributions of TTV are shown for the effects of sex in humans (**A**) and mice (**B**). Data are represented using individual points, boxplot, and half-violin plot. Linear model used to test for sex differences in each species (correcting for age in both species, Euler number in humans, and background strain in mice) ***p<0.0001 . M=male, F=female.

The online version of this article includes the following figure supplement(s) for figure 1:

**Figure supplement 1.** Sex differences in total gray and white matter volume in humans and mice.

Here, we characterize sex differences in global and regional brain volume in a large young adult human cohort from the Human Connectome Project (***Van Essen et al., 2012***) and the young adult mouse cohort (***Ellegood et al., 2015***). In addition to mean differences, we also assess sex differences in the variance of brain volume measures. We quantify the cross-species correspondence of sex-specific brain volume changes in a subset of homologous brain regions based on the directionality and magnitude of volume changes. Furthermore, we build on these anatomical comparisons by assessing whether a brain region's cross-species similarity for sex differences in neuroanatomy is related to its cross-species similarity in expression levels of homologous genes. These quantitative comparisons of neuroanatomical sex differences between humans and mice provide an important reference frame for future studies that seek to use the mouse as a translation model to study sex differences in the human brain. Defining those brain regions with volumetric sex differences that are highly conserved between humans and mice sheds light on evolutionary constraints on sex differences in brain development in mammals and highlights high-priority targets for future translational research.

## Results

### Males have larger brains than females in humans but not in mice

We first examined the effects of sex on total tissue volume (TTV) in humans (597 females/496 males) and mice (213 females/216 males) using structural MRI data from healthy young adults in both species. Replicating a well-established sex differences in prior studies (***Liu et al., 2020***; ***Lotze et al., 2019***; ***Ruigrok et al., 2014***; ***Williams et al., 2021***), we observed in humans that males had significantly larger mean TTV than females (13.5% larger in males; *beta=1.28, t=26.60*, p<2e-16; ***Figure 1A***). In contrast, we did not observe a statistically-significant sex difference in TTV for mice (0.3% larger in males; *beta=0.091, t=0.706*, p=0.481; ***Figure 1B***). In addition to sex differences in TTV, we tested for sex differences in total gray matter and total white matter volume across species to understand whether one tissue type was more implicated in driving the sex difference in volume. In humans, males had significantly larger total gray matter volume (*t=25.36*, p<2e-16) and larger white matter volume (*t=16.02*, p<2e-16) after accounting for age and Euler number (***Figure 1—figure supplement 1A, C***). In contrast, in mice, there was no sex difference in total gray matter volume (*t=0.56*, p=0.68), while males had significantly larger white matter volume (*t=2.01*, p=0.045) after accounting for differences in age and background strain (***Figure 1—figure supplement 1B, D***).

## Sex differences in regional brain volume exist in both humans and mice

After correction for TTV and adjusting for multiple comparisons, we found that 65.8% of human regions showed statistically significant sex differences in volume, of which 63.8% were larger in females and 36.3% were larger in males. In mice, 58.6% of all regions showed statistically significant sex differences in volume, of which 53.0% were larger in females and 47.0% were larger in males. In humans, the median effect size (i.e. standardized beta coefficients) for female-biased (i.e., larger in females) regions was –0.09+/–0.09 standard deviation (SD) (range=−8.90e-16 to –0.44), while it was slightly larger, 0.15+/–0.19 SD (range=0.0008–0.84), for male-biased (i.e., larger in males) regions (*Figure 2A*). In mice, the median effect size across female-biased regions was –0.19+/–0.13 SD (−0.003 to –0.6), similar to that of male-biased regions 0.20+/–0.20 SD (range=0.004–1.04) (*Figure 2B*). Next, we ensured that in humans, the observed sex effects were not influenced by the inclusion of twin or sibling pairs (Appendix 1). Furthermore, we repeated the regional analyses without co-varying for TTV and found that all regional volumes in humans were larger in male due to overall larger brain size. In mice, however, the patterns of sex differences in volume remained largely unchanged likely due to the similarity of total brain size between the sexes (*Figure 2—figure supplement 1*).

In humans, we observed statistically significantly larger regional volume in females than males in the frontal, cingulate, orbital, somatosensory, motor, parietal, parahippocampal, and precuneus cortex, as well as the nucleus accumbens. Males had statistically significantly larger volume in the visual, pareto-occipital, piriform, insula, retrosplenial, medial prefrontal cortex, fusiform face complex, as well as the cerebellum, brainstem, hippocampus, amygdala (including the MeA), the BNST, and hypothalamus (including the MPON) (FDR threshold: t=2.23, q<0.05) (*Figure 2C*). In mice, females had significantly larger regional volumes of the auditory, orbital, entorhinal, anterior cingulate, somatosensory, motor, frontal, and insular cortex, as well as in the caudoputamen and cerebellum. Males had significantly larger regional volume of the olfactory bulb, hippocampus, subiculum, brainstem, amygdala (including MeA), BNST, and hypothalamus (including the MPON) (t=2.23, q<0.05) (*Figure 2D*). For the full list of regional sex differences in human or mouse brain volume, consult ***Source Data Files 1 & 2***, respectively. Notable cross-species congruences in regional volumetric sex differences included larger volume of the frontal, cingulate, orbital, somatosensory, motor, and auditory cortex in females and larger volume of the hypothalamus (including MPON), BNST, amygdala (including MeA), hippocampus, subiculum, brainstem, and cerebellum in males. Notable opposing sex differences between species included the nucleus accumbens, cerebellum, and frontal cortex (male-biased in humans and female-biased in mice) (*Figure 2*). Of note, there are several regions showing a sex-specific volume differences in mice for which we do not have a parcellation/segmentation in the human brain (either due to resolution or lack of atlases), so we cannot conclude that they are mouse-specific.

## Human males have greater variance in brain volume than females, while mice show no sex differences in variance

Next, we evaluated sex differences in the variance of global and regional brain volumes in each species using Levene's test for equality of variances. Variance in TTV was significantly greater in males than females for humans (Levene's test: $F$=10.19, p=0.0014), whereas mice showed no sex difference in TTV variance (Levene's test: $F$=0.765, p=0.382) (*Figure 3AB*). In humans, several regional brain volumes (residualized for TTV, age, and Euler number) showed greater variance in males than females after correction for multiple comparisons (with q<0.05, e.g. posterior parietal, temporal, frontal opercular, medial prefrontal, posterior cingulate, left amygdala, and hypothalamus). In mice, no brain regions (residualized for TTV, age, and background strain) showed sex differences in variance following multiple comparisons correction, however, at a relaxed threshold of p<0.05 mice showed sex differences in regional volumetric variance for *some* regions of the cerebellum, including the culmen, lingula, and fastigial nuclei, as well as in the olfactory bulbs (male >female) and visual, sensorimotor cortex, and CA1 (female >male) (*Figure 3*). These findings remained unchanged after repeating analyses of regional volumetric variance without residualizing for TTV (*Figure 3—figure supplement 1*).

## Sex differences in the size of homologous brain regions are similar across species

We next focused on a set of predefined homologous brain regions (n=60, 28 bilateral and 4 midline) with well-established homology based on comparative studies (*Balsters et al., 2020*; *Beauchamp*

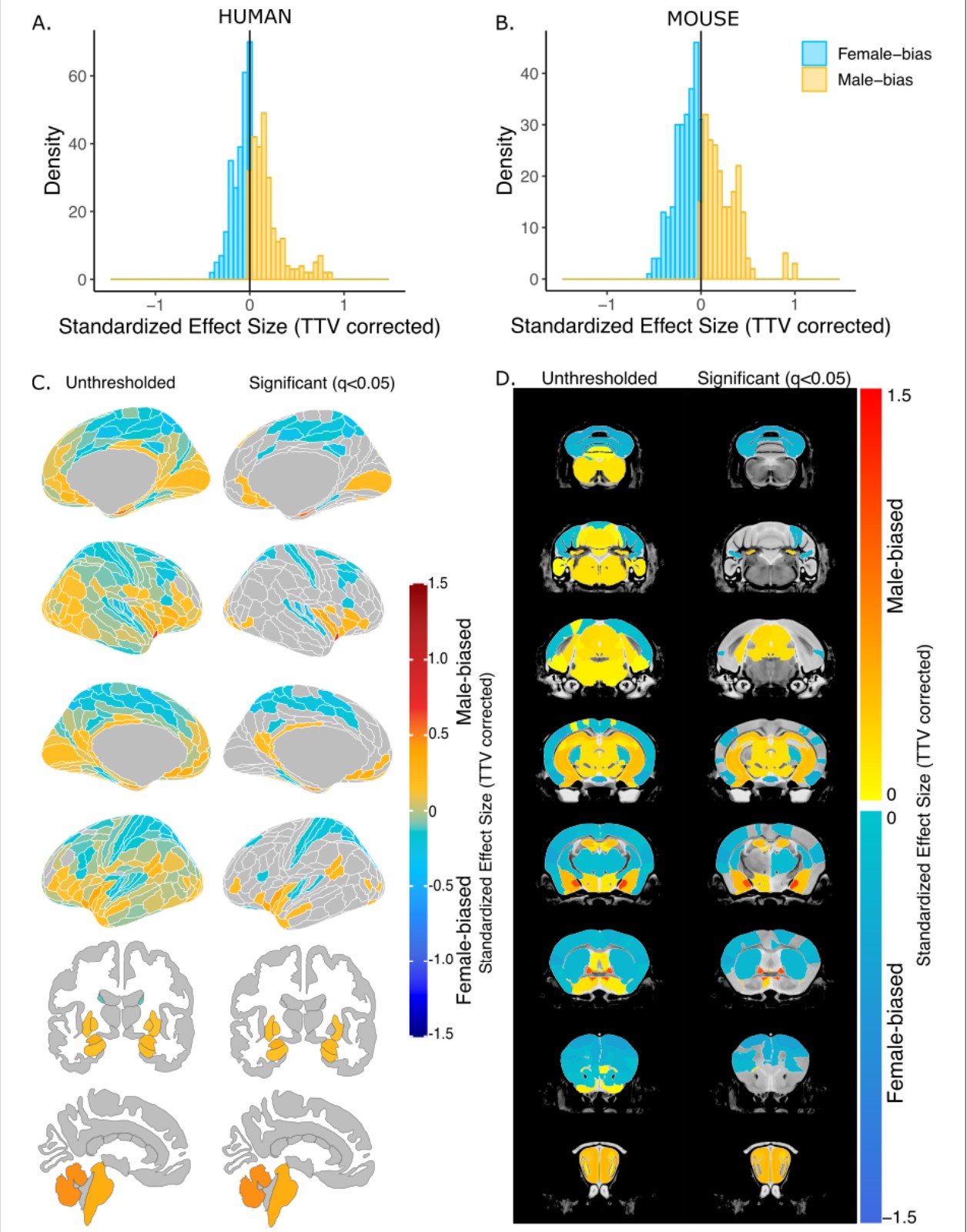

**Figure 2.** Effect of sex on regional brain volume in humans and mice. (**A, B**) Distribution of sex-specific standardized effect sizes across anatomical regions for humans (**A**) and mice (**B**). (**C, D**) Unthresholded (left) and significant (q<0.05; right) standardized effect sizes for the effect of sex displayed on the human (**C**) and mouse (**D**) brains. Regions in yellow-red are larger in males and regions in blue are larger in females; n for humans = 516F/454M, n

*Figure 2 continued on next page*

Figure 2 continued

for mice = 213F/216M. Linear model used to test for sex differences in each species across all regions (correcting for age and TTV in both species, Euler number in humans, and background strain in mice). FDR correction used to identify regions with q<0.05.

The online version of this article includes the following source data and figure supplement(s) for figure 2:

**Source data 1.** Summary of volumetric sex differences across all regions of the human brain.

**Source data 2.** Summary of volumetric sex differences across all regions of the mouse brain.

**Figure supplement 1.** Effect of sex on regional brain volume in humans and mice without total tissue volume (TTV) correction.

*et al., 2022*; *Glasser et al., 2016*; *Gogolla, 2017*; *Swanson and Hof, 2019*; *Vogt and Paxinos, 2014*) to achieve a formal quantitative cross-species comparison of regional sex differences in brain volume. The robust correlation (less sensitive to outliers *Tabatabai et al., 2021*, computed using the *pbcor* R library) of effect size for sex across all homologous brain regions was significant at *r=0.30* (p=0.013) (*Figure 4A*), with a stronger correlation for cortex (*r=0.33, p=0.10*) than non-cortex (*r=0.16, p=0.35*) - although neither of these compartments showed a statistically significant correlation between species in isolation from each other (*Figure 4B*, but note the reduction in sample size in these intra-compartment analyses).

Homologous brain regions that were statistically significantly larger in females (q<0.05) of both species include the bilateral primary somatosensory cortex (human $|\beta|>0.22$, mouse $|\beta|>0.24$), primary auditory cortex (right in humans [$|\beta|>0.16$] and bilateral in mice [$|\beta|>0.20$]), and anterior cingulate cortex (bilateral in humans [$|\beta|>0.10$] and right in mice, $|\beta|>0.20$). The right posterior parietal association area was larger in females of both species but significant (q<0.05) only in humans (human$|\beta|>0.25$, mouse $|\beta|>0.02$), while the bilateral primary motor areas and left thalamus were larger in females of both species but only significant (q<0.05) in mice. Finally, the left thalamus and left ventral orbital area were larger in females but not significant in either species. Homologous regions that were statistically significantly larger in males of both species (q<0.05) include the bilateral amygdala (human $|\beta|>0.20$, mouse $|\beta|>0.15$) (bulk and MeA), bilateral globus pallidus (human $|\beta|>0.14$, mouse $|\beta|>0.11$), hippocampus (human $|\beta|>0.12$, mouse $|\beta|>0.35$; bilateral bulk and CA1, right in humans and bilateral in mice), BNST (human $|\beta|>0.36$, mouse $|\beta|>0.92$), hypothalamus (human $|\beta|>0.62$, mouse $|\beta|>0.11$; bulk and MPON), and brainstem (human $|\beta|>0.35$, mouse $|\beta|>0.20$; medulla and midbrain). Additionally, the left primary visual area, right retrosplenial area, and pons were larger in males in both species but only significant in humans, while CA3 was male-biased in both but only significant in mice (*Table 1*). Regions that showed an incongruent direction of volumetric sex differences between species (with a significant difference in at least one of the species at q<0.05) were the bilateral agranular insula and cerebellar cortex (female-biased in mice, but male-biased in humans) (*Table 1*; *Figure 5*).

As expected, given the robust sex differences for TTV in humans but not mice, we observed a weaker robust correlation for sex differences in the volume of homologous brain regions when repeating the above analyses without controlling for TTV, *r=0.15* (p=0.25) for the effect of sex across homologous regions. Across cortical regions, there was a low correlation of *r=−0.04* (p=0.86), while for non-cortex the correlation was slightly stronger, but negative, *r=−0.20* (p=0.25). In humans, the subset of homologous regions was all larger in males due to the larger overall brain size in males. In mice, we observed the same patterns of sex-bias as we did in the analyses which contrived for TTV except for the right nucleus accumbens showing no sex bias and a male-bias in the pons (both female-biased in the TTV controlled analysis; *Figure 4—figure supplement 1*; *Appendix 2—table 1*).

## Regions that are more congruent between species in their volumetric sex differences tend to be more congruent in their gene expression signatures

Next, we explored whether regions that are more similar between species in their volumetric sex differences are also more similar in their gene expression profile. To derive a region-level measure of between-species congruence in anatomical sex differences (henceforth 'anatomical sex effect similarity score') we multiplied the human and mouse sex effect sizes for each of the 60 homologous brain regions. To derive a region-level measure of cross-species transcriptional similarity, we leveraged gene expression data from the Allen Human and Allen Mouse Brain Atlases (*Hawrylycz et al., 2012*; *Lein et al., 2007*) within the subset of homologous regions defined above. We filtered the gene sets to only

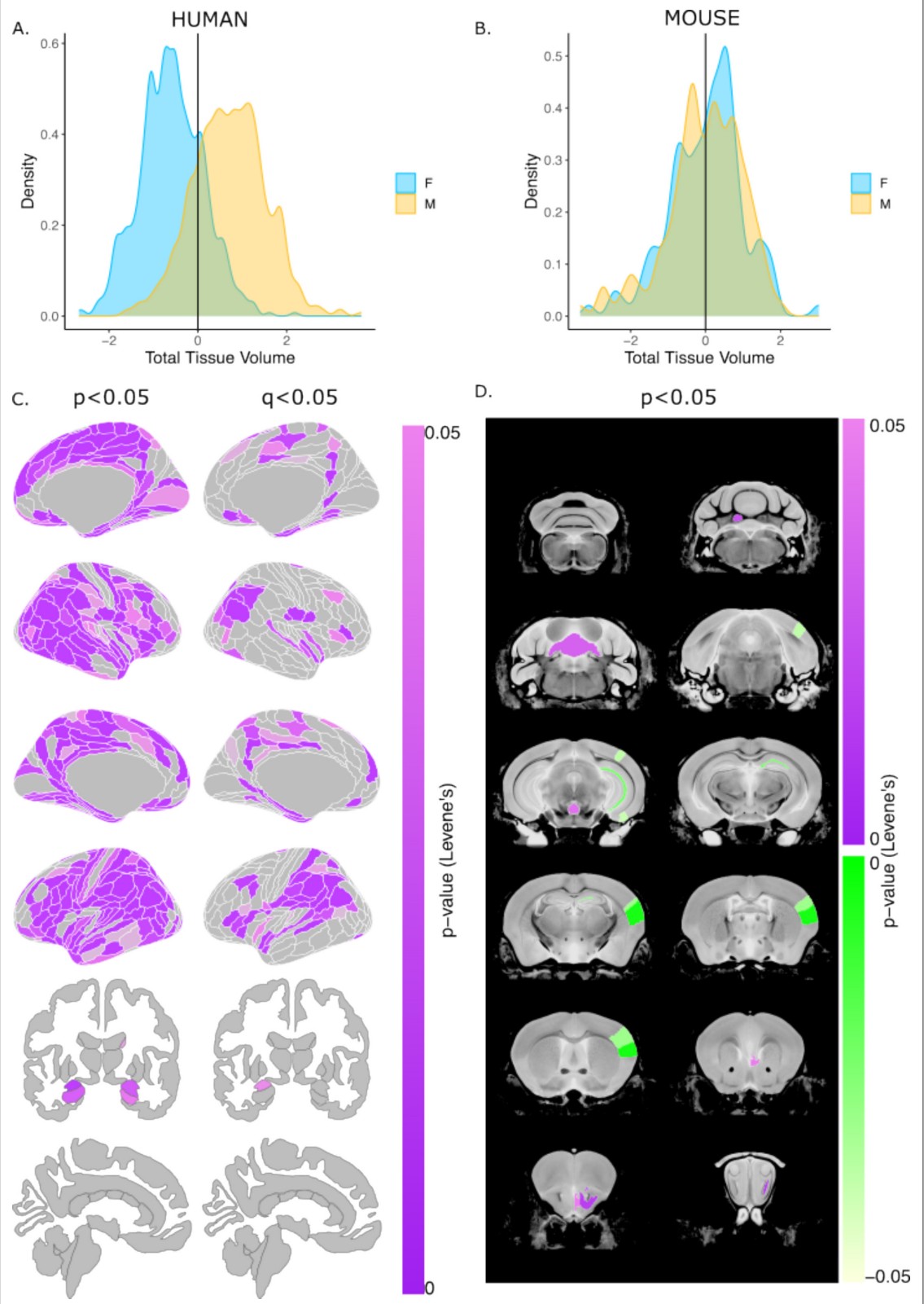

**Figure 3.** Sex differences in the variability of regional brain volumes (accounting for TTV differences) in humans and mice. Distribution of z-scored total brain volume measures across all humans (**A**) and mouse subjects (**B**). Uncorrected (P<0.05; left) and significant (q<0.05; right) sex differences in variability (based on Levene's test) shown on the human brain. Uncorrected (P<0.05) sex differences in variability in the mouse brain (purple = more variable in males; green = more variable in females). Note: all regional human volumes were residualized for TTV, age, and Euler, while regional mouse

*Figure 3 continued on next page*

*Figure 3 continued*

volumes were residualized for TTV, age, and background strain; n for humans = 516F/454M, n for mice = 213F/216M. Levene's test for equality of variances used to test for sex differences in variance (corrected for age and TTV in both species, Euler number in humans, and background strain in mice). P-values were corrected with FDR to derive q-values.

The online version of this article includes the following figure supplement(s) for figure 3:

**Figure supplement 1.** Sex differences in the variability of regional brain volumes (not accounting for TTV differences) in humans and mice.

human-mouse homologous genes (*Beauchamp et al., 2022*; *Coordinators, 2018*) and correlated the regional expression of the homologous genes to derive a measure of transcriptional similarity across species per brain region. These analyses considered the set of 56 homologous brain regions (4 of the original 60 did not have transcriptomic data: bilateral MeA and MPON) and 2835 homologous genes - with supplementary tests based on a priori defined subsets of brain regions and genes (Materials and methods 'Testing if cross-species congruence for sex differences is related to cross-species similarity in gene expression'; Appendix 3). The steps outlined above yielded two measures for each of 56 homologous brain regions: an anatomical sex effect similarity score and a transcriptional similarity score.

The transcriptional similarity score ranged between 0.003 and 0.43 with a mean/median correlation of 0.30, (0.10, interquartile range: 0.25–0.36; *Figure 6—figure supplement 1*). Across homologous brain regions, interregional variation in this transcriptional similarity was positively correlated with anatomical sex effect similarity, *r=0.24* (p=0.08) (*Figure 6A*). Stratification by brain compartment showed that this relationship was stronger, and statistically significant, for cortical regions, *r=0.60* (p=0.0013), than non-cortical regions, *r=0.03* (p=0.88) (*Figure 6B*). Visualizing these relationships indicated that cortical regions showing higher anatomical sex effect similarity scores (mainly primary sensory cortices) tended to show above average transcriptional similarity with each other (i.e. *r*>0.3), whereas cortical regions with lower anatomical sex effect similarity scores (mainly limbic cortex) showed below average transcriptional similarity. Subcortical regions failed to show a correspondence between

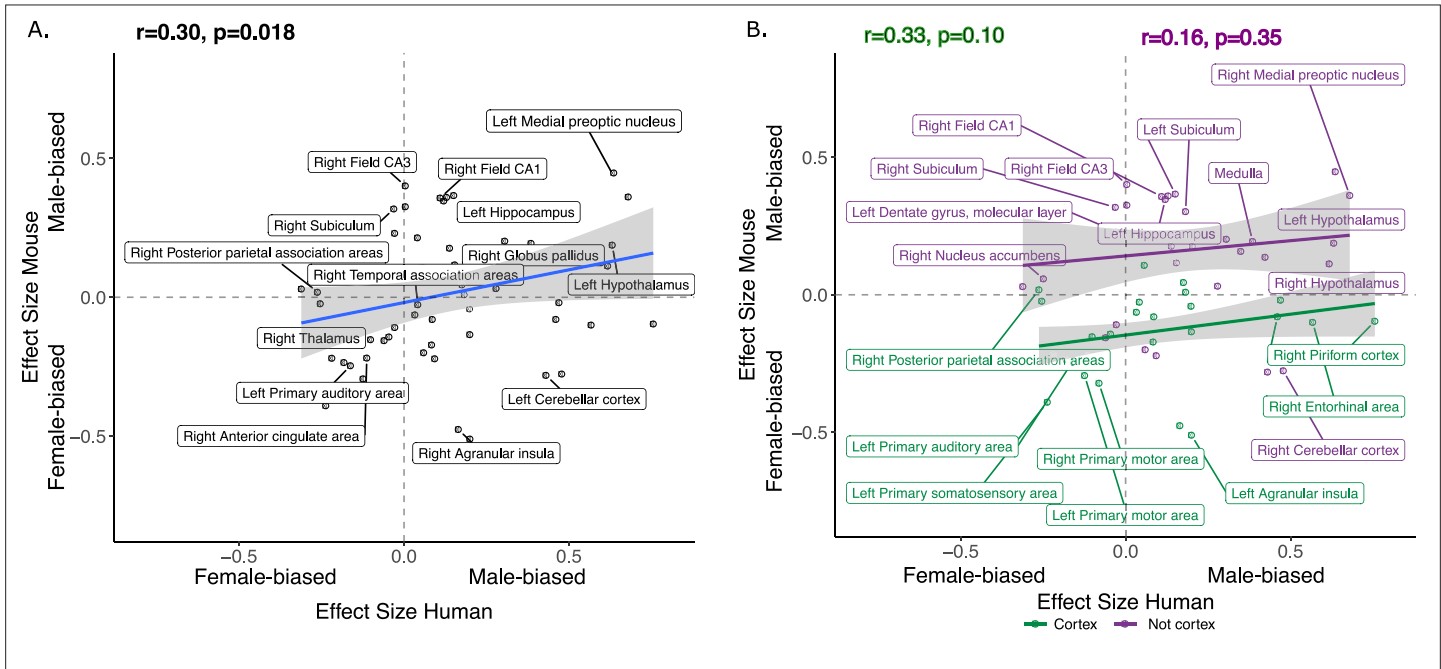

**Figure 4.** Correlation of sex effects on regional volume in homologous regions of the human and murine brain. (**A**) Standardized effect size correlation for the effect of sex in humans (x-axis) and mice (y-axis) (robust correlation coefficient, *r*=0.30). (**B**) Correlation of standardized effect sizes for the effect of sex across species for cortical regions (green, *r*=0.31), and non-cortical regions (purple, *r*=0.16); n for humans = 516F/454M, n for mice = 213F/216M. Robust correlation used to assess correlation between human and mouse sex effect sizes (corrected for TTV and age in both species, Euler number in humans and background strain in mice) yeild r and p values.

The online version of this article includes the following figure supplement(s) for figure 4:

**Figure supplement 1.** Correlation of effects of sex in human and mouse homologous brain regions without total tissue volume (TTV) correction.

**Table 1.** Species-specific effect sizes for volumetric sex differences for 60 homologous brain regions. Effect sizes are color-coded (blue: larger in males/yellow: larger in females) and asterisk/bold text denotes statistical significance. All results are from analyses covarying for total tissue volume (TTV).

| Label | Glasser/Freesurfer# names | Mouse atlas | Hemisphere | Human effect size (β) | Mouse effect size (β) |
|---|---|---|---|---|---|
| Agranular insula | AVI, AAIC, MI | Agranular insular area | L | 0.200 * | –0.558 * |
| | | | R | 0.164 * | –0.522 * |
| Amygdala | Amygdala# | Cortical subplate | L | 0.305 * | 0.163 * |
| | | | R | 0.201 * | 0.151 * |
| Anterior cingulate area | A24pr, a24, p24pr, p24, 24dd, 24dv, p32pr, d32, a32pr, p32, s32 | Anterior cingulate area | L | –0.102 * | –0.198 |
| | | | R | –0.113 * | –0.267 * |
| Bed nucleus of stria terminalis | Bed nucleus of stria terminalis | Bed nucleus of stria terminalis | L | 0.466 * | 0.918 * |
| | | | R | 0.360 * | 0.971 * |
| Caudoputamen | Caudate#, Putamen# | Caudoputamen | L | 0.093 | –0.224 * |
| | | | R | 0.059 | –0.188 * |
| Cerebellar cortex | Cerebellar cortex# | Cerebellar cortex | L | 0.430 * | –0.268 * |
| | | | R | 0.478 * | –0.250 * |
| Dentate gyrus, molecular layer | Dentate gyrus, molecular layer | Dentate gyrus, molecular layer | L | –0.029 | 0.247 * |
| | | | R | 0.041 | 0.232 * |
| CA1 | CA1 | CA1 | L | 0.151 * | 0.385 * |
| | | | R | 0.109 | 0.377 * |
| CA3 | CA3 | CA3 | L | 0.004 | 0.307 * |
| | | | R | 0.004 | 0.411 * |
| Entorhinal cortex | EC | Entorhinal area | L | 0.470 * | –0.090 |
| | | | R | 0.567 * | –0.138 |
| Globus pallidus | Globus Pallidus# | Pallidum | L | 0.154 * | 0.112 * |
| | | | R | 0.138 * | 0.180 * |
| Hippocampus | Hippocampus# | Hippocampal region | L | 0.120 * | 0.353 * |
| | | | R | 0.129 * | 0.379 * |
| Hypothalamus | Hypothalamus | Hypothalamus | L | 0.631 * | 0.185 * |
| | | | R | 0.617 * | 0.109 * |
| Medial amygdalar nucleus | Medial amygdalar nucleus | Medial amygdalar nucleus | L | 0.253 * | 0.906 * |
| | | | R | 0.183 * | 1.034 * |
| Medial preoptic area | Medial preoptic area | Medial preoptic area | L | 0.636 * | 0.435 * |
| | | | R | 0.680 * | 0.367 * |
| Nucleus accumbens | Nucleus accumbens# | Striatum ventral region | L | –0.311 * | –0.005 |
| | | | R | –0.249 * | 0.032 |
| Perirhinal area | PeEc, TF, PHA2, PHA3 | Perirhinal area | L | 0.033 | –0.120 |
| | | | R | 0.086 | –0.108 |

*Table 1 continued on next page*

*Table 1 continued*

| Label | Glasser/Freesurfer# names | Mouse atlas | Hemisphere | Human effect size (β) | Mouse effect size (β) |
|---|---|---|---|---|---|
| Piriform cortex | Pir | Piriform cortex | L | 0.460 | –0.131 |
| | | | R | **0.756 *** | –0.151 |
| Posterior parietal association areas | 5 m, 5 mv, 5 L | Posterior parietal association areas | L | **–0.254 *** | 0.016 |
| | | | R | **–0.263 *** | 0.039 |
| Primary auditory area | A1 | Primary auditory area | L | –0.163 | **–0.256 *** |
| | | | R | **–0.182 *** | **–0.209 *** |
| Primary motor area | 4 | Primary motor area | L | –0.124 | **–0.329 *** |
| | | | R | –0.081 | **–0.357 *** |
| Primary somatosensory area | 1, 2, 3 a, 3b | Primary somatosensory area | L | **–0.237 *** | **–0.419 *** |
| | | | R | **–0.219 *** | **–0.241 *** |
| Primary visual area | V1 | Primary visual area | L | **0.175 *** | 0.029 |
| | | | R | **0.199 *** | –0.102 |
| Retrosplenial area | RSC | Retrosplenial area | L | **0.198 *** | 0.004 |
| | | | R | **0.182 *** | 0.035 |
| Subiculum | PreS | Subiculum | L | **0.182 *** | **0.317 *** |
| | | | R | –0.031 | **0.338 *** |
| Temporal association areas | FFC, PIT, TE1a, TE1p, TE2a, TF, STV, STSvp, STSva | Temporal association areas | L | 0.057 | 0.085 |
| | | | R | 0.042 | –0.004 |
| Thalamus | Thalamus# | Thalamus | L | –0.028 | –0.098 |
| | | | R | –0.060 | **–0.139 *** |
| Ventral orbital area | 10 r, 10 v | Ventral orbital area | L | 0.083 | **–0.209 *** |
| | | | R | –0.046 | –0.171 |
| Brain stem (midline) | Brainstem# | Midbrain, Hindbrain | M | **0.349 *** | **0.200 *** |
| Medulla (midline) | Medulla | Medulla | M | **0.385 *** | **0.204 *** |
| Midbrain (midline) | Midbrain | Midbrain | M | **0.423 *** | **0.191 *** |
| Pons (midline) | Pons | Pons | M | **0.279 *** | 0.065 |

anatomical sex effect similarity and transcriptional similarity - but we noted that this could be driven by the influences of the BNST as an outlier region (Cook's d: left BNST=0.20, right BNST=0.13). However, the correlation between anatomical sex effect similarity and transcriptional similarity remained low for subcortical regions after exclusion of the BNST from analysis (*r=0.02*, p=0.92) (***Figure 6—figure supplement 2***).

Finally, we asked if the observed correlations between anatomical sex effect similarity and transcriptional similarity would be significantly modified by recomputing correlations using biologically informed subsets of homologous genes: (i) X-linked genes (n=91) or (ii) sex hormone genes (n=34). Across these sensitivity analyses, we observed a similar correlation when using X-linked genes (*r*=0.25, p=0.07) compared to the full gene set with stronger correlations in cortical (*r*=0.62, p=0.0007) vs. non-cortical (*r*=0.30, p=0.11) regions (Appendix 3). Interestingly, we observed an exception to this pattern for the subset of sex hormone genes involved in androgen vs. estrogen and progesterone pathways. Transcriptional similarity scores based on androgen pathways were weakly correlated with anatomical sex effect similarity scores in the cortex (*r*=0.05, p=0.81), but were strongly correlated with anatomical sex effect similarity in the non-cortical regions (*r*=0.46, p=0.01). In contrast, the correlation between

| | Female-biased in Human | Male-biased in Human |
|---|---|---|
| **Male-biased in Mouse** | Posterior Parietal Association Cortex<br>Nucleus Accumbens (R)<br>Dentate Gyrus (L)<br>Subiculum (R) | **Amygdala * (medial amygdala *)**<br>**Hypothalamus* (MPON *)**<br>**Hippocampus * (**R DG, **CA1 *,** CA3**)**<br>**Globus Pallidus *, BNST ***<br>**Brainstem* (**Pons, **Midbrain *, Medulla *)**<br>Retrosplenial Cortex<br>Subiculum (L)<br>Temporal Association Cortex (L) |
| **Female-biased in Mouse** | **Anterior cingulate cortex ***<br>**Primary Auditory Cortex ***<br>**Primary Somatosensory Cortex ***<br>Primary Motor Cortex<br>Thalamus<br>Nucleus Accumbens (L)<br>Posterior Parietal Association Cortex (L)<br>Ventral Orbital Cortex (R) | **Agranular Insula ***<br>**Cerebellar Cortex ***<br>Caudoputamen<br>Entorhinal Cortex<br>Perirhinal Cortex<br>Piriform Cortex<br>Primary Visual Cortex<br>Temporal Association Cortex (R)<br>Ventral Orbital Cortex (L) |

**Figure 5.** Homologous brain regions show either congruent or divergent sex bias across species. There are no regions that are larger in human females and mouse males (top left quadrant, light green). Several regions show male-bias (i.e., larger volume in males; top right quadrant in yellow) and female-bias (i.e., larger volume in males; females; bottom left quadrant in blue) across species. A subset of regions shows male-bias in humans but female-bias in mice (bottom right quadrant in green). * denotes significant sex effect in both species; n for humans = 516F/454M, n for mice = 213F/216M. Ordering of quadrants matches the quadrants of the scatter plots in *Figure 3* and *Figure 5*.

anatomical sex effect similarity and transcriptional similarity based on estrogen/progesterone pathways was positive in the cortex ($r$=0.29, p=0.15) but negative in the non-cortex ($r$=−0.27, p=0.15) (*Appendix 3—figure 1*). For combined sex hormone analysis see *Appendix 3—figure 1*. Finally, we recomputed the transcriptional similarity by randomly resampling various subsets of homologous genes 10,000 times, and then correlated those similarity values to the anatomical sex congruence across regions (see *Appendix 3—figure 2*). Gene lists and available in *Source data 1* while GO terms can be found in *Source data 2*.

## Discussion

This study provides the first cross-species comparison of the effects of sex on human and mouse brain anatomy. Our findings suggest that sex differences in overall brain volume are not conserved across species, but that there is a meaningful degree of cross-species concordance for sex differences in regional volumes. Furthermore, regions with more similar sex effects between species also tended to show more similar transcriptional profiles - particularly amongst cortical brain regions. This work has consequences for understanding sex differences in the mammalian brain, the use of mice as translational models for human sex differences, and the broader topic of comparative structural neuroimaging between humans and mice.

First, in line with many previous observations (*Liu et al., 2020*; *Lotze et al., 2019*; *Ruigrok et al., 2014*; *Williams et al., 2021*), we find that in humans, males have larger mean total brain volume than females. In contrast, we observed no sex differences in total brain size in mice, in line with a recent study reporting no sex differences in total brain volume, cell density, and total cell number (*Elkind et al., 2023*). In previous studies, only subtle sex differences in total brain size have been

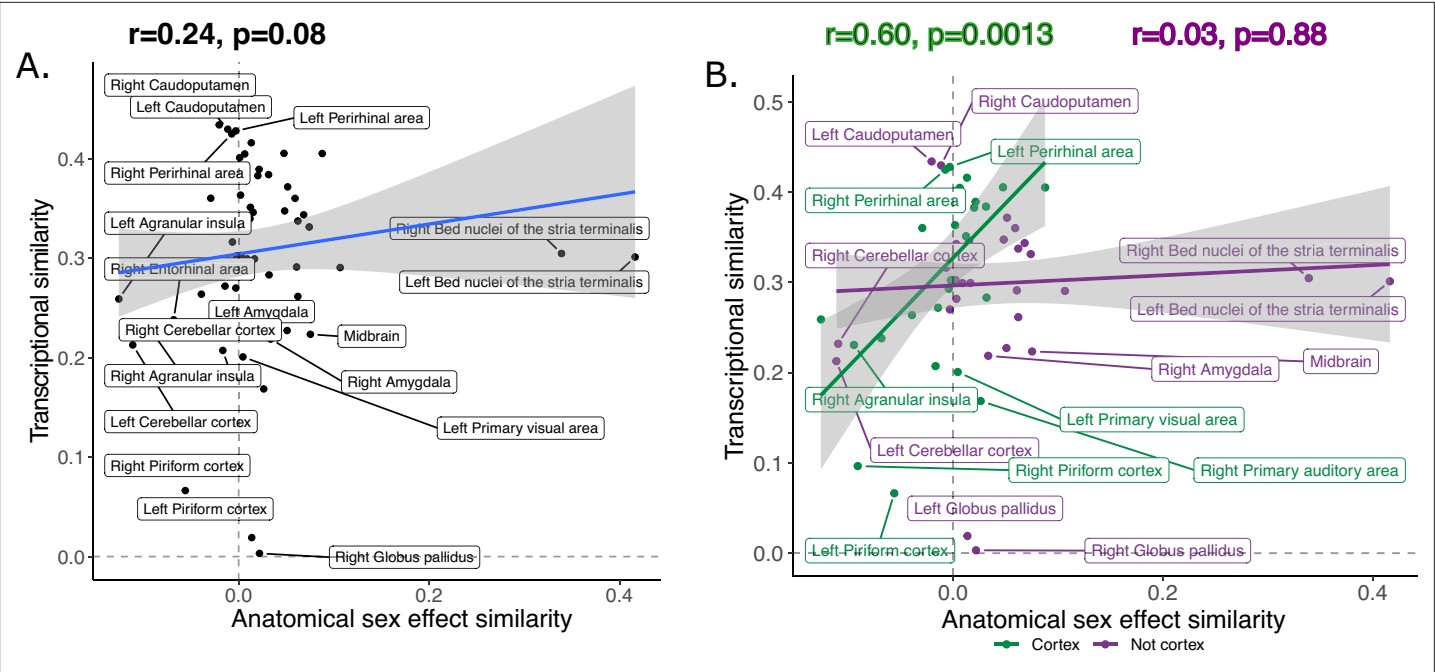

**Figure 6.** Inter-species anatomical sex congruence and gene expression shows modest correlation across homologous brain regions. Correlation between similarity in volumetric sex differences and similarity in transcriptional profile using all homologous genes across all homologous regions is modest (**A**).There was a stronger correlation across cortical (green) compared to non-cortical (purple) regions (**B**). Robust correlation used to assess correlation between anatomical sex effect similarity and transcriptional similarity yeild r and p values.

The online version of this article includes the following figure supplement(s) for figure 6:

**Figure supplement 1.** Similarity matrix for cross-species homologous gene expression.

**Figure supplement 2.** Recomputing inter-species anatomical sex congruence and gene expression exlcuding BNST shows similar correlation to analysis including all homologous brain regions.

reported in C57BL6/J mice (2.5% larger male brain). However, this subtle difference in brain size is lost when accounting for sex differences in body weight (**Spring et al., 2007**), which is not the case in humans where the sex difference in brain size is dampened but not lost when accounting for body size (**Dekaban, 1978**; **Williams et al., 2021**).

As seen for sex differences in total brain volume, humans and mice also showed a striking contrast for sex differences in anatomical variance. In humans, males showed more variance for both total and regional brain volume measures, in line with previous findings (**Forde et al., 2020**). In mice, no sex differences in anatomical variance were observed. While variance in brain anatomy has not been previously studied in mice, greater male variability has been observed for morphological traits, while greater female variability has been observed for immunological and metabolic traits (**Zajitschek et al., 2020**). The observation of greater neuroanatomical variance in human, but not mouse males questions the theory that males are more variable because they are heterogametic (have XY chromosomes) (**Reinhold and Engqvist, 2013**), since male mice are also heterogametic. Gaining an understanding of the causes of sex differences in the variability of brain structures may provide clues for understanding male-specific brain development or neurodevelopmental disorders (**Wierenga et al., 2022**).

After accounting for global brain size differences, we identified several sex-specific differences in regional volume in the human and mouse brain, several of which showed a consistent sex difference across species. For these regions, available mechanistic information in mice can be used to refine mechanistic hypotheses in humans. We find that males of both species have larger amygdala (including the MeA), hippocampus, BNST, and hypothalamus (including the MPON) volumes in line with a previous human (**Lotze et al., 2019**; **Neudorfer et al., 2020**; **Ruigrok et al., 2014**) and mouse neuroimaging studies (**Qiu et al., 2018**). Rodent studies have shown that the emergence of sMRI-defined volume differences aligns with the developmental timing of sex differences in apoptosis in the BNST and MPON, (**Chung et al., 2000**), and sex differences in the synaptic organization in the MeA

(*Cooke and Woolley, 2005*; *Nishizuka and Arai, 1981*); these male-biased regions also tend to show greater density than female-biased brain regions (*Elkind et al., 2023*). To further support the organizational role of hormones, masculinized female mice display male-typical sex differences in the BNST and MeA (i.e., larger volume) as well as in behavior (*McCarthy, 2020*; *Wu and Shah, 2011*). Neuroimaging studies of FCG mice identified independent effects of sex hormones from sex chromosome dosage on brain anatomy. For example, the MeA is larger in mice with testes than mice with ovaries, but is smaller in XY than XX mice, which further points to the importance of sex hormones in sculpting this specific brain region (*Corre et al., 2016*; *Vousden et al., 2018*). While we do not have causal mechanistic data in humans, neuroimaging studies have shown that male-biased regional volume differences emerge in early development, and may be sensitive to the amount of fetal testosterone exposure (*Knickmeyer et al., 2014*; *Lombardo et al., 2012*). In females, we show that both species have a larger anterior cingulate, somatosensory, and primary auditory cortex in line with human (*Lotze et al., 2019*; *Neudorfer et al., 2020*; *Ruigrok et al., 2014*) and mouse studies (*Qiu et al., 2018*). While the mechanisms driving female-bias in regional brain volume are not as well understood, neuroimaging studies in both species find that female-specific volume enlargements tend to emerge during puberty (*Knickmeyer et al., 2014*; *Qiu et al., 2018*), which may point to an important role of pubertal hormones in shaping brain structure. Regions showing congruent sex difference across species, such as the primary somatosensory cortex, may be high-priority targets for conducting mechanistic studies in the mouse brain that have relevance to the human.

We also observe regions where there is a significant sex difference in volume in one species that is absent or inverted in the other, including the agranular insula and cerebellar cortex (male-biased in humans and female-biased in mice). This incongruence may be due to species differences in the composition of these regions. For example, a recent study comparing the cerebellum of humans, macaques, and mice identified a group of progenitor cells present in the human brain but not in the mouse or macaque (*Haldipur et al., 2019*). Furthermore, a comparison of cell populations in the middle temporal gyrus of humans and mice found that in humans, there is greater interaction between neurons and non-neuronal cells as well as greater diversity and density of glia, which may in turn lead to cross-species differences in brain size (*Fang et al., 2022*). The divergence in sex effects may be due to a divergence in gene expression patterns. This idea is supported by our analyses that incorporate information on bulk regional gene expression, which is heavily shaped by cellular composition (*Yao et al., 2021*). Brain regions with more similar gene expression profiles between species (and likely more similar cellular compositions) also tend to show more similar volumetric sex differences across species. Finally, the lack of congruence in the cross-species sex effects may also be due to species differences in the function of the brain regions.

Our comparison of homologous brain regions based on the similarity of anatomical sex differences and homologous gene expression revealed good cross-species alignment between those metrics. This relationship was much stronger in the cortical areas than in non-cortical areas. Within cortical regions, the primary sensory areas showed stronger transcriptional and anatomical sex congruence, while limbic structures showed lower congruence. In non-cortical regions, the transcriptional similarity was consistent across regions, independent of anatomical sex congruence. Limiting the homologous gene set of X-linked genes did not alter the congruence between the anatomical effects and transcriptional similarity, however, limiting the genes to include sex hormone genes did. The transcriptional similarity based on androgen genes was weakly correlated with cortical sex effects but strongly correlated with non-cortical sex differences, while the transcriptional similarity based on estrogen and progesterone genes was positively correlated with cortical but negatively correlated with non-cortical sex effects. This interesting observation may indicate a differential role for sex hormones in patterning either cortical or non-cortical sex differences in the brain, which warrants further investigation.

The findings presented here indicate that there is modest cross-species alignment for regional sex differences, but that humans and mice show very different effects of sex on overall brain size. Interpretation of this cross-species difference is challenged by our limited causal understanding of the mechanisms driving sex differences in overall brain size within humans. It has been theorized that a potential evolutionary driver for sex differences in body size (and the contributions of this to brain size) is sexual selection for larger body size in species where there is higher competition for mates (*Plavcan, 2012*). However, evolutionary causes for inter-species variation in the sex effects on total brain volume are hard to test empirically. In contrast, there is empirical evidence for the role

of sex hormones and sex chromosomes in shaping sex differences in brain size, therefore, species differences in the modulation of these biological processes may contribute to species differences in brain size (*McCarthy and Arnold, 2011*). Large-scale neuroimaging studies in humans suggest that sex differences in total brain volume are already apparent in toddlerhood (*Bethlehem et al., 2022*), at birth (*Dean et al., 2018*; *Gilmore et al., 2007*), and even prenatally (*Conte et al., 2018*; *Griffiths et al., 2023*). These differences are, therefore, likely to reflect pre- and perinatal influences of sex differences in circulating sex steroids and/or sex chromosome dosage on the human brain, where males are exposed to a perinatal testosterone surge and females are not (*McCarthy, 2020*). While sex differences in the total brain size of mice have not been well characterized in prenatal and early postnatal life, there is some experimental in vitro evidence of differential gonadal steroid effects on early neurogenesis in humans vs. mice. In human brain organoids (both XX and XY) in vitro androgen exposure (as a model for the perinatal 'mini-puberty' seen in humans *Kelava et al., 2022*) has been associated with an increased proliferation of excitatory cortical progenitors and radial glia, increasing the neurogenic niche (*Kelava et al., 2022*). In contrast, in mouse brain organoids, estradiol, but not testosterone exposure leads to an increase in progenitor cell proliferation (*Kelava et al., 2022*). This species difference aligns with evidence that gonadal steroids masculinize the mouse brain via activation of the estrogen E2 receptor by aromatized testosterone, whereas the masculinizing effects of gonadal steroids on the primate brain are more dependent on direct activation of the androgen receptor by testosterone (*Schwarz and McCarthy, 2008*; *Zuloaga et al., 2008*). There is also some evidence that sex chromosome dosage may be differentially related to brain size in humans vs. mice. In humans with sex chromosome aneuploidy, increased dosage of Y-chromosomes is associated with increased total brain size, while increased X-chromosome dosage is associated with decreased total brain size (*Guma et al., 2023*; *Raznahan et al., 2016*). Importantly, these global differences are not recapitulated in the mouse brain, where extra X- or Y-chromosome dosage is not associated with any differences in brain size (*Guma et al., 2023*). This may be, in part, explained by species differences in the size and gene content of the Y-chromosome; mice have an expanded proportion of ampliconic genes compared to primates (*Soh et al., 2014*), as well as in the process of XCI, where more genes escape inactivation in humans (12–15%) than mice (2–5%) (*Deng et al., 2014*).

The work presented here should be considered in light of some caveats and limitations. First, while we used detailed segmentation to characterize sex differences in the human and mouse brain, several canonically sex-biased nuclei in mice are difficult to detect with human MRI. Second, as with our previous study (*Guma et al., 2023*), we employ a parcellation-based approach which is limited to anatomically defined brain structures for which one-to-one mapping across species may not always be accurate (*Mars et al., 2018*). Third, we focus on the comparison of brain anatomy in young adulthood, but extending across different spatial and temporal scales would be critical to our understanding of the emergence and evolution of sex differences across species. Fourth, our incorporation of gene expression data relied on available atlases that are largely (humans) or exclusively (mice) derived from male brains. Fifth, while the comparison of humans and mice provides an important first step in comparative analyses of sex-biased brain development in mammals, the formal phylogenetic analysis will require the incorporation of data from other animals including non-human primates. Lastly, although we carefully characterize sex differences in brain anatomy, it is important to stress that such differences provide no information regarding brain function or behavior (*DeCasien et al., 2022*). Moreover, sex differences in human brain development are likely to be uniquely shaped by the closely related but distinct construct of gender, which is typically considered to be absent in all non-human animals including mice.

Notwithstanding these limitations and caveats, we show that sex differences in global brain size are not well conserved across species, but that sex differences in regional brain volume do show some congruence between humans and mice. Furthermore, we find that human-mouse congruence in volumetric sex differences of a subset of homologous brain regions is stronger for cortical as compared to non-cortical structures, and - especially in the cortex - echoed by cross-species similarities in regional gene expression. In conclusion, this quantitative comparison of sex-biased neuroanatomy across humans and mice may inform future translational studies aimed at using mice to better understand sex differences in the human brain.

**Table 2.** Demographics for human sample.

|  |  | Females | Males | Statistics |
|---|---|---|---|---|
| Sample size |  | 516 | 454 |  |
| Age | *Mean* | 29.41 | 27.9 | $F_{(1,1082)}=57.13$, $p=8.67e-14$ *** |
|  | *SD* | 3.68 | 3.61 |  |
|  | *Range* | 22–36 | 22–37 |  |
| Education (in years) | *Mean* | 14.96 | 14.84 | $F_{(1,1082)}=1.465$, $p=0.226$ |
|  | *SD* | 1.83 | 1.77 |  |
|  | *Range* | 11–17 | 11–17 |  |
| Euler number | *Mean* | –52.47 | –58.26 | $F_{(1,1082)}=25.72$, $p=4.65e-07$ *** |
|  | *SD* | 17.66 | 19.25 |  |
|  | *Range* | –126 to –16 | –136 to –16 |  |
| Zygosity | *Monozygotic* | 102 | 49 | $X^2=26.281$, $p=8.33e-06$ |
|  | *Dizygotic* | 163 | 131 |  |
|  | *Not Twin* | 251 | 274 |  |

*p<0.01 **p<0.001 for ANOVA test of significant difference between groups (males vs. females). SD=standard deviation.

## Materials and methods

### Human participants and neuroimaging data

#### Data acquisition

The human sample included 3T T1-weighted 0.7 mm³ sMRIs from healthy young adults (597 females/496 males aged 22–35 years) from the Human Connectome Project (HCP) 1200 release. Recruitment procedures and scan acquisition parameters (T1-MPRAGE: TR=2400 ms; TE=2.14 ms; TI=1000 ms; Flip angle=8 deg; FoV=224 × 224 mm) are detailed in the original publication. All individuals provided informed consent to participate in the study (*Van Essen et al., 2012*; *Glasser et al., 2013*). HCP data were provided (in part) by the Washington University – the University of Minnesota Consortium of the Human Connectome projects (principal investigators: David Van Essen and Kamil Ugurbil: 1U54MH091657). For more information about applying to get access to the HCP restricted data and for the HCP restricted data use terms see: https://www.humanconnectome.org/study/hcp-young-adult/document/wu-minn-hcp-consortium-restricted-data-use-terms. Participant characteristics are detailed in *Table 2*.

#### Data processing

#### Cortical morphometry

T1-weighted sMRI data were preprocessed using the PreFreesurfer pipeline, described in detail here (*Glasser et al., 2013*). Next, we used Freesurfer's (version 7.1.0) (*Fischl, 2012*) *recon-all* with the *highres* flag to reconstruct and parcellate the cortex of all individuals at the original resolution of the data (*Zaretskaya et al., 2018*). This pipeline is freely available for download, documented (http://surfer.nmr.mgh.harvard.edu/), and well described in previous publications (*Dale et al., 1999*; *Dale and Sereno, 1993*; *Desikan et al., 2006*; *Fischl et al., 2004a*; *Fischl et al., 2002*; *Fischl et al., 2001*; *Fischl et al., 1999*; *Fischl et al., 1998*; *Fischl and Dale, 2000*; *Han et al., 2006*; *Jovicich et al., 2006*; *Kuperberg et al., 2003*; *Reuter et al., 2010*). The *mri_anatomical_stats* utility was used to extract several features including cortical volume from the cortical surfaces. These vertex-level measures were averaged across 360 regions from the multimodally informed Glasser Human Connectome Project atlas (*Glasser et al., 2016*).

## Subcortical morphometry

For subcortical segmentation, each voxel is assigned one of 43 labels using the Freesurfer 'aseg' feature (version 7.1.0; see *Fischl et al., 2004b*; *Fischl et al., 2002* for full details). Of the 40 labels, 20 were included in analyses as they segmented gray matter structures. We aimed to build upon the standard segmentations above to include more nuclei previously shown to be sex-biased. We used FreeSurfer's joint segmentation of hippocampal subfields (*Iglesias et al., 2015a*), sub-nuclei of the amygdala (*Saygin et al., 2017*), and brainstem (*Iglesias et al., 2015b*). Since the hypothalamus and related nuclei including the BNST are not available through FreeSurfer, we used a different published atlas (*Neudorfer et al., 2020*) (https://zenodo.org/record/3942115). Segmentation was performed by registering the atlas labels to our study-specific average (using deformation-based morphometry processing with ANTs-based tools https://github.com/CoBrALab/optimized_antsMultivariateTemplateConstruction (*Devenyi, 2024a*), Appendix 4). Voxel-wise volume differences were summed within the regions of interest of the hypothalamic atlas to generate structure volumes. Finally, each segmentation (cortical and subcortical) was visually quality controlled to ensure region boundaries matched anatomy, and excluded if there was a segmentation fault. Additionally, participants with an Euler number (extracted from each individual's cortical reconstruction) less than −200 were excluded from statistical analyses based on previous reports (*Rosen et al., 2018*).

## Mouse subjects and neuroimaging data

### Data acquisition

Mice included in this study were all wild-type controls from a large collection of autism mouse models made up of separate cohorts from diverse labs, sent to the Mouse Imaging Centre in Toronto for neuroimaging (*Ellegood et al., 2015*). To model normative sex differences in the young adult mouse brain, we included wild-type mice from lab cohorts that had a minimum of five males and five females (to allow for appropriate covariation of potential background genotype and strain effects) on a C57BL6 (J or N) background strain. This yielded n=216 males (mean age=postnatal day [PND] 62.7+/−8.5; range=PND 56–90) and n=213 females (mean age=PND 62.0+/−7.5; range=PND 56–90). We harmonized neuroimaging measures between cohorts within background strain (134 F/141 M C57BL6J mice from 12 cohorts, 79 F/75 M C57BL6N mice from 6 cohorts, *Table 3* & Appendix 5) using ComBat from the *sva* library in R. This method is a popular adjustment method initially developed for genomics data (*Johnson et al., 2007*), but adapted to neuroimaging harmonization to harmonize measurements across scanners (*Fortin et al., 2018*; *Fortin et al., 2017*).

Young adult mice were transcardially perfused following a standard protocol which was consistent across mouse cohorts (*Cahill et al., 2012*). Fixed brains (kept in the skull to avoid distortions) were scanned at the Mouse Imaging Centre on a multichannel 7.0 T scanner with a 40 cm diameter bore magnet (Varian Inc, Palo Alto, CA). A T2-weighted fast spin echo sequence was used with the following scan parameters: T2W 3D FSE cylindrical *k*-space acquisition sequence, TR/TE/ETL=350 ms/12 ms/6, two averages, FOV/matrix-size=20×20×25 mm/ 504 × 504 × 630, total-imaging-time = 14 hr (*Spencer Noakes et al., 2017*). All procedures were approved by the animal care committees of the originating labs. The data presented here was compliant with all ethical regulations concerning animal experimentation and was approved by the animal care committee at The Centre for Phenogenomics (AUP-0260H) at the University of Toronto and all the other institutions that provided animals.

### Data processing

sMRIs were registered using an unbiased deformation-based morphometry registration pipeline (*Avants et al., 2011*; *Avants et al., 2009*; *Collins et al., 1994*; *Eskildsen et al., 2012*; *Friedel et al., 2014*). This results in an average brain, from which log-transformed Jacobian determinants can be calculated (*Chung et al., 2001*); these encode voxel-wise volume differences between each individual mouse brain and the average brain. sMRIs were also segmented into 355 unique brain regions using previously published atlases (*Dorr et al., 2008*; *Richards et al., 2011*; *Steadman et al., 2014*; *Ullmann et al., 2013*) with the MAGeT brain algorithm (*Chakravarty et al., 2013*; *Pipitone et al., 2014*). Visual quality control was performed to evaluate the accuracy of registrations and segmentations.

**Table 3.** Demographics for mouse sample.
For details regarding the origin of each mouse cohort refer to *Appendix 5—table 1*.

| | Female | Male | Statistics |
|---|---|---|---|
| Sample size | 213 | 216 | |
| **Age** | | | |
| Mean | 62.0 | 62.8 | |
| SD | 7.5 | 8.6 | |
| Range | 56–90 | 56–90 | $F(1,70)=0.78$, $p=0.38$ |
| **Background Strain** | | | |
| C57BL-6J | 134 | 141 | |
| C57BL-6N | 79 | 75 | |
| **Mouse Cohort for C57BL6J** | | | |
| A | 10 | 12 | |
| B | 15 | 15 | |
| C | 27 | 29 | |
| D | 13 | 7 | |
| E | 8 | 11 | |
| F | 9 | 10 | |
| G | 7 | 9 | |
| H | 10 | 10 | |
| I | 7 | 9 | |
| J | 10 | 6 | |
| K | 9 | 8 | $X^2=5.46$, $p=0.91$ |
| L | 9 | 15 | |
| **Mouse Cohort for C57BL6N** | | | |
| M | 13 | 19 | |
| N | 10 | 8 | |
| O | 25 | 13 | |
| P | 9 | 9 | |
| Q | 9 | 13 | |
| R | 13 | 12 | $X^2=5.745$, $p=0.332$ |

## Gene expression data

### Human gene expression data

Human gene expression data were obtained from the Allen Human Brain Atlas (*Hawrylycz et al., 2012*), downloaded from the Allen Institute's API (http://api.brain-map.org), and preprocessed using *abagen* package in Python (https://abagen.readthedocs.io/en/stable/) (*Arnatkeviciute et al., 2019*; *Hawrylycz et al., 2012*; *Markello et al., 2021*) as previously described by *Beauchamp et al., 2022*. Data from all six donors was preprocessed as described in *Beauchamp et al., 2022* yielding a gene-by-sample expression matrix with 15,627 genes and 3702 samples across all donors.

## Mouse gene expression data

Mouse gene expression data were obtained from the Allen Mouse Brain Atlas (*Lein et al., 2007*). Briefly, the whole-brain in-situ hybridization expression data were downloaded using Allen Institute's

API (http://help.brain-map.org/display/api/Downloading+3-D+Expression+Grid+Data) coronal in-situ hybridization experiments and reshaped into 3D images in the Medical Image NetCDF (MINC) format and preprocessed as previously described by *Beauchamp et al., 2022*. The result of this pre-processing pipeline was a gene-by-voxel expression matrix with 3958 genes and 61,315 voxels.

## Expression matrices for homologous genes within homologous regions

To obtain gene expression data for each homologous brain region described above, the human atlases used to perform the segmentations were registered into MNI ICBM 152 2009 c non-linear symmetric space (*Fonov et al., 2011*; *Fonov et al., 2009*), and each human sample was annotated with one of the 60 (28 bilateral and 4 midline) brain region labels. Several regions were missing expression data from the human right hemisphere, including the right hypothalamus, BNST, DG, and CA3, so we reflected the left hemisphere data to the right hemisphere to increase our sample for subsequent analyses. Furthermore, the MeA and MPON were missing transcriptomic data and were excluded from these analyses, yielding a total of 56 homologous brain regions. Similarly, the mouse atlas was registered into the Allen Mouse Brain Common Coordinate Framework (CCFv3) space (*Wang et al., 2020*) and each voxel was annotated with one of the 56 brain region labels. Next, we created a gene-by-region expression matrix for the human and mouse. For the human, we weighted the average of gene expression data based on the volume of the region of interest; this was particularly important since we combined several regions within the human atlases to obtain our homologous regions. For the mice, we averaged voxel-level gene expression data within regions of interest. To obtain homologous genes, each species' full gene set was intersected with a list of 3331 homologous genes obtained from the NCBI HomoloGene database (*Coordinators, 2018*), yielding 2835 homologous genes present in both species' expression matrices, as described by *Beauchamp et al., 2022*. Each matrix was z-scored across brain regions to normalize gene expression measures.

## Statistical analysis
### Sex differences in mean global and regional brain volume
#### Human

All statistical analyses were performed in R version 3.4.2. A linear model was used to test for the effect of sex ($\beta_1$: male vs. female) on z-scored global or regional brain volumes, with mean-centered age ($\beta_2$), TTV ($\beta_3$), and Euler number ($\beta_4$) as covariates ($\varepsilon$: error term). The beta-value for the effect of sex ($\beta_1$) is referred to as a standardized effect size as it was computed on standardized (z-scored) volumes. The false discovery rate (FDR) correction (*Benjamini and Hochberg, 1995*; *Benjamini and Yekutieli, 2001*) was applied to control for multiple comparisons with the expected proportion of false positives(q) set to 0.05. The formula, using 'ROI volume' as the example region:

ROI volume ~intercept + $\beta_1$(Sex: male vs. female) + $\beta_2$(age−mean age) + $\beta_3$(TTV) + $\beta_4$(Euler number) + $\varepsilon$

To ensure that the effects we observed were not driven by the inclusion of twin or sibling pairs, we re-ran the same model on a subset of the data that randomly included only one of the two twin pairs. Additionally, we ran a linear-mixed effects model on the full data set with family ID as a random intercept. We correlated the standardized effect size for sex from both of these models to the ones generated from our main model to ensure that the effects were equivalent (Appendix 1).

#### Mouse

The analysis performed in humans was replicated in mice by using a linear model to test for the effect of sex ($\beta_1$: male vs. female) on z-scored global or regional brain volume. Mean-centered age ($\beta_2$), TTV ($\beta_3$), and background strain ($\beta_4$) were also included as covariates ($\varepsilon$: error term). Again, the beta-value for sex ($\beta_1$) is referred to as a standardized effect size.

ROI volume ~intercept + $\beta_1$(Sex: male vs. female) + $\beta_2$(age−mean age) + $\beta_3$(TTV) + $\beta_4$(Background Strain) + $\varepsilon$

In both species, we repeated the regional analyses described above without covarying for TTV. Finally, we used a Levene's test to assess sex differences in variance of global and regional brain volume in each species.

## Cross-species comparison

### Sex-biased brain anatomy

To test for the convergence between sex effects in humans and mice, we considered a subset of 60 brain regions for which there is well-established homology based on comparative structural and functional studies (*Balsters et al., 2020*; *Beauchamp et al., 2022*; *Glasser et al., 2016*; *Gogolla, 2017*; *Swanson and Hof, 2019*; *Vogt and Paxinos, 2014*). We leveraged previous work which maps brain regions between humans and mice using six cytoarchitectonic and MRI-derived human atlases, and three cytoarchitectonic mouse atlases (as well as two rat atlases) in order to narrow down regions with cross-species homologs (*Swanson and Hof, 2019*). We computed the standardized effect sizes of sex on the volume of each of these regions as described in the section above for humans and mice. We used a robust correlation to determine the similarity of sex differences in regional brain volume across species by correlating the effect size for sex between species across regions. We also repeated this analysis using effect size estimates derived in each species without covarying for TTV.

### Testing if cross-species congruence for sex differences is related to cross-species similarity in gene expression

To derive a regional measure for the cross-species concordance of sex differences in volume we multiplied the effect size of sex for each species. This product (i.e. 'anatomical sex effect similarity score') is positive for regions showing congruent sex differences between species (i.e. larger in males for both or larger in females for both), and negative for regions showing incongruent sex effects (e.g. larger in males for females for humans but larger in males for mice). Next, to derive a regional measure of cross-species transcriptional similarity, we computed the Pearson correlation between scaled expression values for all homologous genes in humans and mice per brain region. These steps created two measures per brain region - one for the cross-species similarity of volumetric sex differences (anatomical sex effect similarity) and another for the cross-species similarity in gene expression - which we could then correlate across brain regions to test if regions with more conserved sex differences show more conserved gene expression. We estimated this correlation using a robust correlation - once using measures from all brain regions and all genes and again using subsets of brain regions and genes. Specifically, we compared the correlation between conserved sex effects and conserved expression for cortical vs. subcortical regions and recomputed these correlations using the following subsets of homologous genes: (i) X-linked genes (n=91) or (ii) sex hormone signaling genes (n=34). Next, we split the sex hormone genes into either (i) androgen-signaling related genes (n=11), or (ii) estrogen or progesterone-signaling related genes (n=23). The sex hormone signaling genes were identified based on Gene Ontology data for biological process modules from the Bader Lab (University of Toronto, http://baderlab.org/GeneSets). For each analysis using subsets of genes, we generated a null distribution of correlations based on 10,000 randomly sampled gene sets of the same size (i.e. 91 to match X-chromosome genes), and compared the observed correlations with these null distributions using the 'get_p_value' function in R's infer package (Appendix 3).

## Acknowledgements

This study was supported by the intramural research program of the National Institute of Mental Health (project funding: 1ZIAMH002949-03), the National Institute of Child Health and Disease (R01HD100298), as well as Canadian Institutes of Health Research, BrainCanada, and the Ontario Brain Institute. EG also receives salary support from the Fonds de Recherche du Québec en Santé. This research was enabled in part by support provided by Compute Canada (https://computecanada.ca/). We would also like to thank Dr. Yohan Yee for his contributions to the gene annotations.

# Additional information

## Funding

| Funder | Grant reference number | Author |
|---|---|---|
| Eunice Kennedy Shriver National Institute of Child Health & Human Development | R01HD100298 | Armin Raznahan |
| National Institute of Mental Health | 1ZIAMH002949-03 | Armin Raznahan |
| Fonds de Recherche du Québec - Santé | | Elisa Guma |
| Canadian Institutes of Health Research | | Jason P Lerch |
| Brain Canada | | Jason P Lerch |
| Ontario Brain Institute | | Jason P Lerch |

The funders had no role in study design, data collection and interpretation, or the decision to submit the work for publication.

## Author contributions

Elisa Guma, Conceptualization, Data curation, Formal analysis, Investigation, Visualization, Methodology, Writing – original draft, Writing – review and editing; Antoine Beauchamp, Data curation, Formal analysis, Investigation, Visualization, Methodology, Writing – review and editing; Siyuan Liu, Data curation, Formal analysis, Methodology, Writing – review and editing; Elizabeth Levitis, Linh Pham, Data curation, Formal analysis, Writing – review and editing; Jacob Ellegood, Data curation, Methodology, Writing – review and editing; Rogier B Mars, Conceptualization, Methodology, Writing – review and editing; Armin Raznahan, Conceptualization, Resources, Funding acquisition, Investigation, Methodology, Writing – original draft, Project administration, Writing – review and editing; Jason P Lerch, Conceptualization, Data curation, Supervision, Funding acquisition, Investigation, Methodology, Project administration, Writing – review and editing

## Author ORCIDs

Elisa Guma https://orcid.org/0000-0003-4651-8529
Siyuan Liu http://orcid.org/0000-0003-3661-6248
Armin Raznahan https://orcid.org/0000-0002-5622-1190

## Ethics

Recruitment procedures scan acquisition parameters (T1-MPRAGE: TR=2400ms; TE=2.14ms; TI=1000ms; Flip angle=8 deg; FoV=224mm) are detailed in the original publication; all individuals provided informed consent to participate in the study (Van Essen et al., 2012; Van Essen et al., 2013). HCP data were provided (in part) by the Washington University - University of Minnesota Consortium of the Human Connectome projects (principal investigators: David Van Essen and Kamil Ugurbil: 1U54MH091657). For more information about applying to get access to the HCP restricted data and for the HCP restricted data use terms see: https://www.humanconnectome.org/study/hcp-young-adult/document/wu-minn-hcp-consortium-restricted-data-use-terms.

The data presented here was compliant with all ethical regulations concerning animal experimentation and was approved by the animal care committee at The Centre for Phenogenomics (AUP-0260H) at the University of Toronto and all the other institutions that provided animals.

Joint Public Review: https://doi.org/10.7554/eLife.92200.2.sa1

## Additional files

### Supplementary files

• MDAR checklist

• Source data 1. Sex hormone gene and sex chromosome lists for both species. Gene list based on gene ontology search terms (from Bader Lab) used to identify genes associated with sex hormones including androgen, estrogen, and progesterone, as well as X-chromosome genes.

• Source data 2. Homologous sex hormone and sex chromosome genes. Genes identified in *Source data 1* were filtered to only include homologous genes to allow for cross-species comparison.

### Data availability

Further information and requests for resources should be directed to and will be fulfilled by the lead contacts, Dr. Armin Raznahan (raznahana@mail.nih.gov) and Dr. Jason Lerch (jason.lerch@ndcn.ox. ac.uk). Input regional volume measures for humans and mice, as well as original code for statistical analyses have been deposited to GitHub and are publicly available here: https://github.com/elisa-guma/Normative-Sex-Differences (copy archived at *Guma, 2024*), while code for gene expression processing is available here: https://github.com/abeaucha/NormativeSexDifferences (copy archived at *Beauchamp, 2024*). For more information about applying to get access to the HCP restricted data and for the HCP restricted data use terms see: https://www.humanconnectome.org/study/hcp-young-adult/document/wu-minn-hcp-consortium-restricted-data-use-terms.

The following previously published datasets were used:

| Author(s) | Year | Dataset title | Dataset URL | Database and Identifier |
|---|---|---|---|---|
| Van Essen et al | 2013 | 1200 Subjects Data Release | https://www.humanconnectome.org/study/hcp-young-adult/document/1200-subjects-data-release | The Human Connectome Project, 1200-subjects-data-release |
| Allen Institute for Brain Science | 2010 | Human Brain Atlas Microarray | https://knowledge.brain-map.org/data/X9PGJ220940UEDBLHP0/summary | Allen Human Brain Atlas, X9PGJ220940UEDBLHP0 |
| Allen Institute for Brain Science | 2004 | Allen Mouse Brain Atlas ISH | https://knowledge.brain-map.org/data/KC4VT2XQDVRD6KUVNOE/summary | Allen Institute for Brain Science, KC4VT2XQDVRD6KUVNOE |

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

## Appendix 1

### Ensuring there is no bias due to relatedness of twin pairs

To ensure that the relatedness of some of the human subjects did not impact our estimations of sex-differences in brain anatomy, we recomputed our standardized effect sizes in two ways. First, we randomly removed one of the twin pairs from the sample and recomputed the effect size for sex by running the same linear model as described in the main text (see 'Sex differences in mean global and regional brain volume: Human'), which included the main effects of sex, mean-centered age, TTV, and Euler number ($\beta_4$). Second, we ran a linear mixed-effects model on the full sample testing again for effects of sex, mean-centered age, TTV, and Euler number, and accounted for relatedness by modeling a family ID as the random intercept. From both models, we extracted the standardized effect size (beta-coefficient) for the main effect of sex and correlated with the effect size for the main effect of sex generated from our main model ('Sex differences in mean global and regional brain volume: Human').

### Comparison of sex-biased maps when accounting for relatedness in the human sample

Sex-differences in regional brain volume were not affected by the relatedness of twin pairs; both unthresholded and thresholded maps for the effect of sex are identical across our three different models (*Appendix 1—figure 1A, B*). The random exclusion of one twin pair yielded regional effect size estimates for sex that were highly correlated to those generated from the model that did not exclude a twin pair (*r*=0.955, t=228.25, df = 489, p-value<2.2e-16). Similarly, regional effect size estimates for sex generated from the linear mixed-effects model that accounted for relatedness were highly correlated to those generated from the linear model that did not account for relatedness (*r*=0.993, t=190.35, df = 490, p-value<2.2e-16) (*Appendix 1—figure 1C, D*).

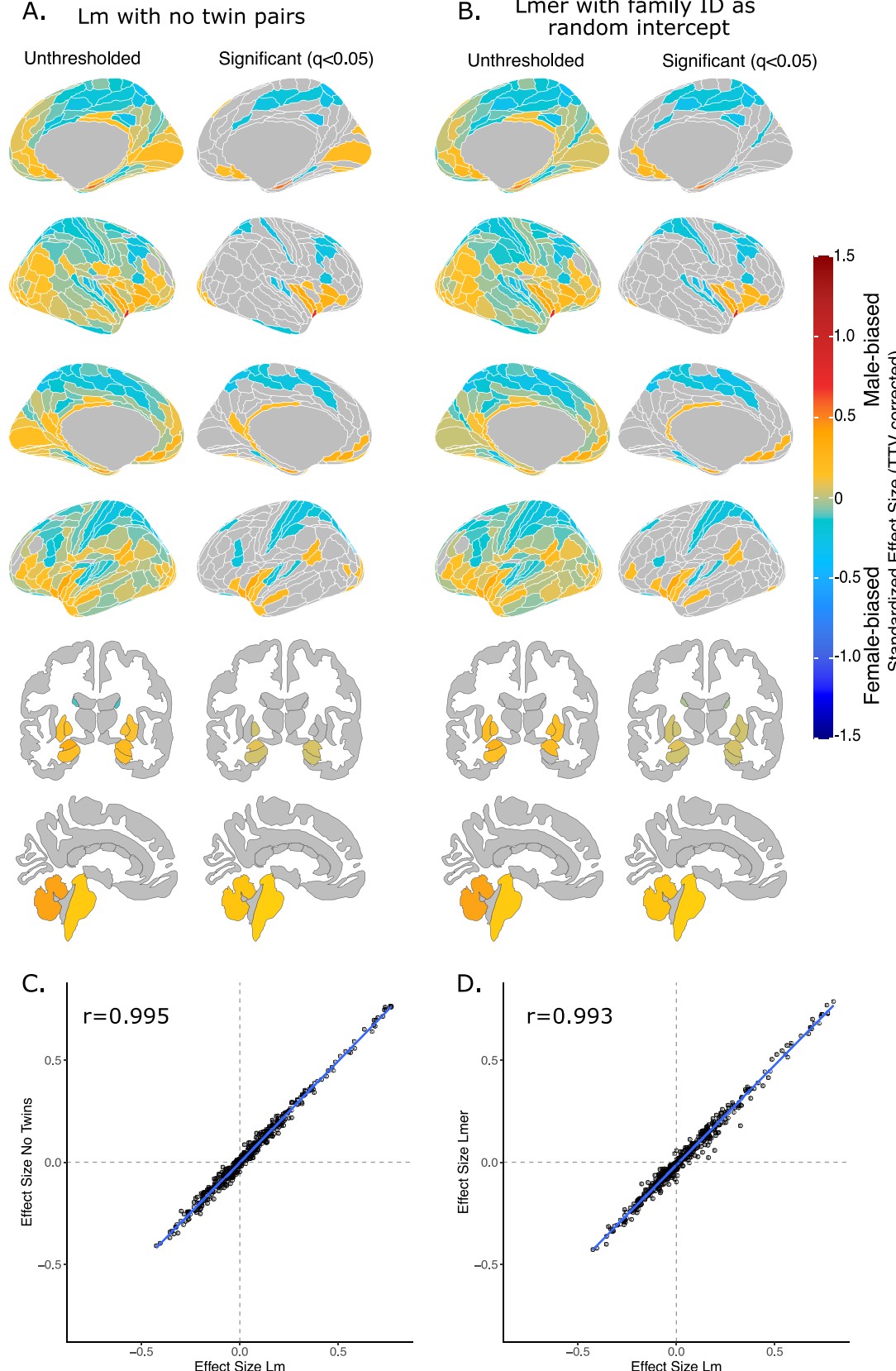

**Appendix 1—figure 1.** Effect of sex on regional brain volume in humans when accounting for relatedness. Distribution of sex-biased standardized effect sizes for humans when one twin-pair is excluded (**A**) or when *Appendix 1—figure 1 continued on next page*

*Appendix 1—figure 1 continued*

relatedness is accounted for using linear mixed-effects modeling (**B**). Unthresholded (left) and significant (q<0.05; right) standardized effect sizes for the effect of sex displayed on the human brains. Regions in yellow-red are male-biased and regions in blue are female-biased. (**C**) Pearson correlation of standardized effect size for sex generated from a linear model with no twin pairs with effect size for sex generated from linear model with twin pairs included (r=0.995). (**C**) Pearson correlation of standardized effect size for sex generated from a linear model including twin pairs with effect size for sex generated from a the linear mixed-effects model which included twin pairs, but accounting for relatedness by using a family ID as a random effect. Human sample size excluding twin pairs: n=412F/403M; sample size including twin pairs: n=516F/454M.

## Appendix 2

**Appendix 2—table 1.** Mapping of homologous human-mouse brain regions (not TTV-corrected standardized effect sizes).
Blue shade highlights female-biased regions, while yellow shade highlights male-biased regions. In humans, the subset of homologous regions was all male-biased due to the larger overall brain size in males. In mice, we observed the same patterns of sex-bias as we did in the analyses which contrived for total tissue volume (TTV) except for the right nucleus accumbens showing no sex bias and a male-bias in the pons (both female-biased in the TTV controlled analysis).

| Label | Glasser/Freesurfer# names | Mouse atlas | Hemisphere | Human effect size | Mouse effect size |
|---|---|---|---|---|---|
| Agranular insula | AVI, AAIC, MI | Agranular insular area | L | 1.025 * | –0.484 * |
| | | | R | 0.985 * | –0.453 * |
| Amygdala | Amygdala# | Cortical subplate | L | 1.194 * | 0.265 * |
| | | | R | 1.135 * | 0.240 * |
| Anterior cingulate area | A24pr, a24, p24pr, p24, 24dd, 24dv, p32pr, d32, a32pr, p32, s32 | Anterior cingulate area | L | 0.946 * | –0.107 |
| | | | R | 0.958 * | –0.173 |
| Bed nucleus of stria terminalis | Bed nucleus of stria terminalis | Bed nucleus of stria terminalis | L | 1.160 * | 0.937 * |
| | | | R | 1.131 * | 0.959 * |
| Caudoputamen | Caudate#, Putamen# | Caudoputamen | L | 0.986 * | –0.158 |
| | | | R | 0.993 * | –0.132 |
| Cerebellar cortex | Cerebellar cortex# | Cerebellar cortex | L | 1.123 * | –0.220 * |
| | | | R | 1.192 * | –0.218 |
| Dentate gyrus, molecular layer | Dentate gyrus, molecular layer | Dentate gyrus, molecular layer | L | 0.885 * | 0.292 * |
| | | | R | 0.935 * | 0.275 * |
| CA1 | CA1 | CA1 | L | 0.936 * | 0.427 * |
| | | | R | 0.925 * | 0.417 * |
| CA3 | CA3 | CA3 | L | 0.593 * | 0.382 * |
| | | | R | 0.663 * | 0.460 * |
| Entorhinal cortex | EC | Entorhinal area | L | 0.925 * | 0.038 |
| | | | R | 0.984 * | –0.047 |
| Globus pallidus | Globus Pallidus# | Pallidum | L | 1.040 * | 0.167 * |
| | | | R | 1.049 * | 0.229 * |
| Hippocampus | Hippocampus# | Hippocampal region | L | 1.045 * | 0.414 * |
| | | | R | 1.033 * | 0.426 * |
| Hypothalamus | Hypothalamus | Hypothalamus | L | 1.374 * | 0.245 * |
| | | | R | 1.354 * | 0.170 |
| Medial amygdalar nucleus | Medial amygdalar nucleus | Medial amygdalar nucleus | L | 0.547 * | 0.926 * |
| | | | R | 0.654 * | 1.057 * |
| Medial preoptic area | Medial preoptic area | Medial preoptic area | L | 1.205 * | 0.472 * |
| | | | R | 1.275 * | 0.388 * |

*Appendix 2—table 1 Continued on next page*

*Appendix 2—table 1 Continued*

| Label | Glasser/Freesurfer# names | Mouse atlas | Hemisphere | Human effect size | Mouse effect size |
|---|---|---|---|---|---|
| Nucleus accumbens | Nucleus accumbens# | Striatum ventral region | L | 0.596 * | 0.079 |
| | | | R | 0.709 * | 0.106 |
| Perirhinal area | PeEc, TF, PHA2, PHA3 | Perirhinal area | L | 0.839 * | –0.014 |
| | | | R | 0.821 * | –0.029 |
| Piriform cortex | Pir | Piriform cortex | L | 0.988 * | –0.020 |
| | | | R | 1.076 * | –0.039 |
| Posterior parietal association areas | 5 m, 5 mv, 5 L | Posterior parietal association areas | L | 0.519 * | 0.028 |
| | | | R | 0.554 * | 0.066 |
| Primary auditory area | A1 | Primary auditory area | L | 0.352 * | –0.192 |
| | | | R | 0.357 * | –0.183 |
| Primary motor area | 4 | Primary motor area | L | 0.785 * | –0.237 * |
| | | | R | 0.795 * | –0.269 * |
| Primary somatosensory area | 1, 2, 3 a, 3b | Primary somatosensory area | L | 0.743 * | –0.332 * |
| | | | R | 0.671 * | –0.162 |
| Primary visual area | V1 | Primary visual area | L | 0.729 * | 0.098 |
| | | | R | 0.703 * | –0.085 |
| Retrosplenial area | RSC | Retrosplenial area | L | 0.838 * | 0.010 |
| | | | R | 0.770 * | 0.062 |
| Subiculum | PreS | Subiculum | L | 0.643 * | 0.370 * |
| | | | R | 0.433 * | 0.382 * |
| Temporal association areas | FFC, PIT, TE1a, TE1p, TE2a, TF, STV, STSvp, STSva | Temporal association areas | L | 1.084 * | 0.159 |
| | | | R | 1.060 * | 0.024 |
| Thalamus | Thalamus# | Thalamus | L | 1.019 * | –0.040 |
| | | | R | 0.994 * | –0.087 |
| Ventral orbital area | 10 r, 10 v | Ventral orbital area | L | 0.666 * | –0.121 |
| | | | R | 0.605 * | –0.093 |
| Brain stem (midline) | Brainstem# | Midbrain, Hindbrain | M | 1.209 * | 0.226 * |
| Medulla (midline) | Medulla | Medulla | M | 1.111 * | 0.255 * |
| Midbrain (midline) | Midbrain | Midbrain | M | 1.280 * | 0.194 * |
| Pons (midline) | Pons | Pons | M | 1.101 * | 0.101 |

## Appendix 3

### Correlating the anatomical sex effect similarity score to the similarity of homologous gene expression across homologous brain regions

We limited our homologous genes to those that were only expressed on the X-chromosome to determine whether their regional expression would be more highly correlated to patterns of sex-biased neuroanatomy. The robust correlation between anatomical sex similarity score and transcriptional similarity of X-chromosome genes (n=91) was slightly stronger, although very similar to that of the full homologous gene set (n=2835), *r=0.25* (p=0.07). As with the full gene set, the relationship was stronger, and statistically significant, for cortical regions, *r=0.62* (p=0.0007), than non-cortical regions, *r=0.30* (p=0.11).

Filtering genes associated with sex steroid hormones based on Gene Ontology annotations (n=30) yielded a weak, and negative correlation, *r=−0.11* (p=0.44). Interestingly, the relationship for cortical regions was positive and similar to the relationships described above, *r=0.29*, (p=0.14), while the correlation in the subcortex was negative, *r=−0.13* (p=0.50) (*Source data 1* for gene lists with GO terms and *Source data 2* for those lists filtered to include only homologous genes). Lastly, we examined whether limiting sex hormone genes to more male-biased (androgen signaling genes, n=11) or more female-biased (estrogen and progesterone signaling genes, n=23) genes affected the correlation between anatomical sex similarity score and gene expression similarity. We found that androgen genes were positively correlated (*r=0.23*, p=0.08) with the anatomical sex similarity score, with a weak correlation for cortex (*r=0.05*, p=0.81) but a strong one for non-cortex (*r=0.46*, p=0.01). In contrast, estrogen and progesterone genes were negatively correlated with the anatomical sex similarity score (*r=−0.21*, p=0.13), with a positive correlation in the cortex (*r=0.29*, p=0.15) and a negative relationship in the non-cortex (*r=−0.27*, p=0.15).

Finally, we recomputed the transcriptional similarity by randomly resampling various subsets of homologous genes 10,000 times, and then correlated those similarity values to the anatomical sex congruence across regions. The subsets were chosen to reflect the number of genes selected for the various hypothesis-driven subsets, i.e., n=91 for X-chromosome genes, n=34 for sex hormone genes, n=11 for androgen genes, and n=23 for estrogen and progesterone genes. For the distribution based on 91 genes, we observed a mean r of 0.05 (min = −0.20, max = 0.32); for 34 genes we observed a mean r of 0.04 (min = −0.34, max = 0.39); for 11 genes we observed a mean r of 0.03 (min = −0.42, max = 0.46), and for 23 genes we observed a mean r of 0.04 (min = −0.33, max = 0.38). Relative to the null distribution of correlated values, the relationship anatomical sex congruence and transcriptional similarity were significant for the subset of X-chromosome genes (n=91, $p_{null}$ = 0.0014), trending towards significant for the subset of sex hormone genes (n=34, $p_{null}$ = 0.08), and for androgen genes (n=11, $p_{null}$ = 0.06), and significant for the subset of estrogen and progesterone genes (n=23, $p_{null}$ = 0.01).

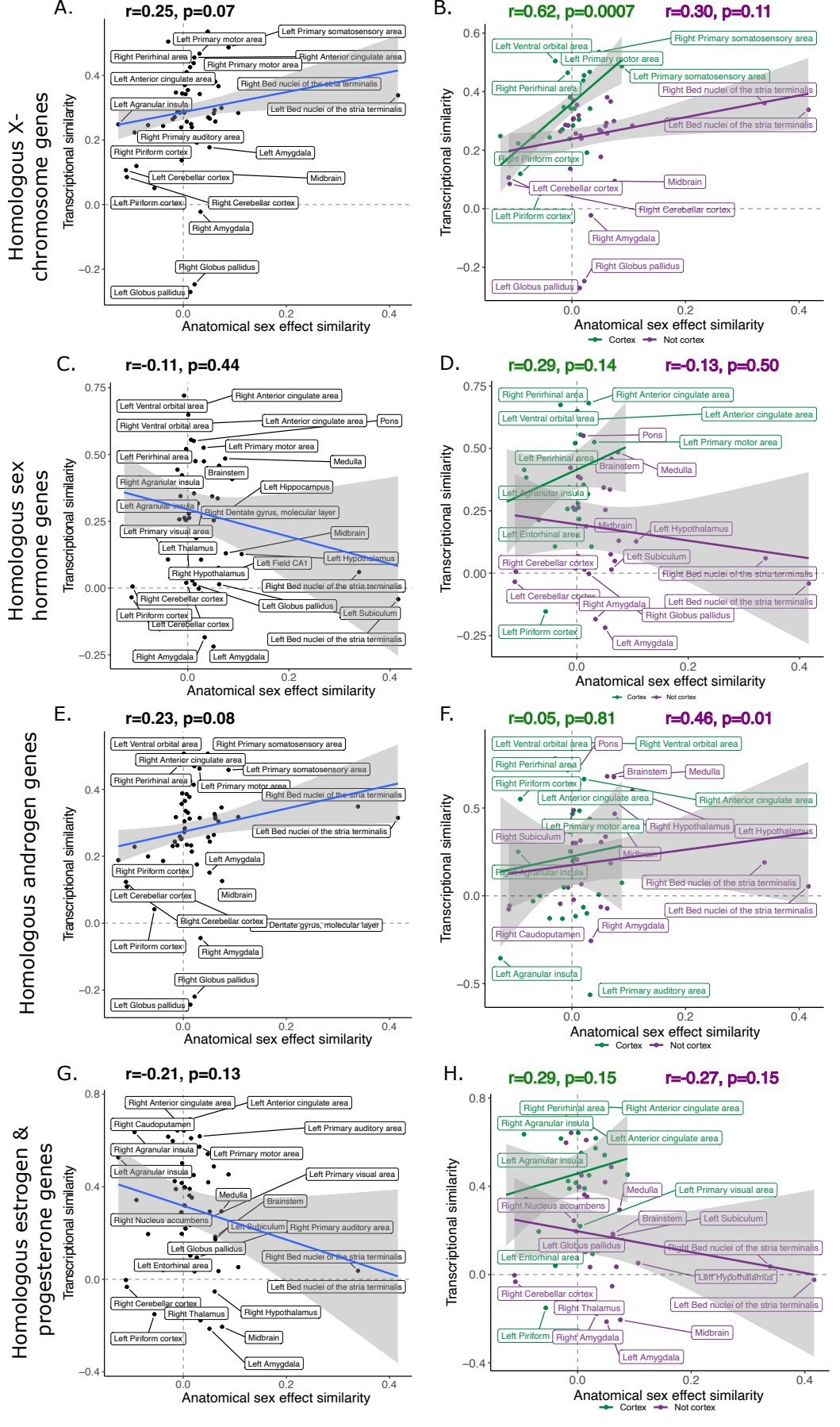

**Appendix 3—figure 1.** Robust correlation of anatomical similarity of sex effects and transcriptional similarity of homologous brain regions across species. Robust correlation of anatomical similarity and transcriptional similarity using only X-chromosome homologous genes (n=91; **A, B**), using only sex hormone genes androgen, estrogen, progesterone genes (n=30; **C, D**), just androgen genes (**E, F**), or just estrogen and progesterone genes (**G, H**) across all homologous regions or split into cortical and non-cortical.

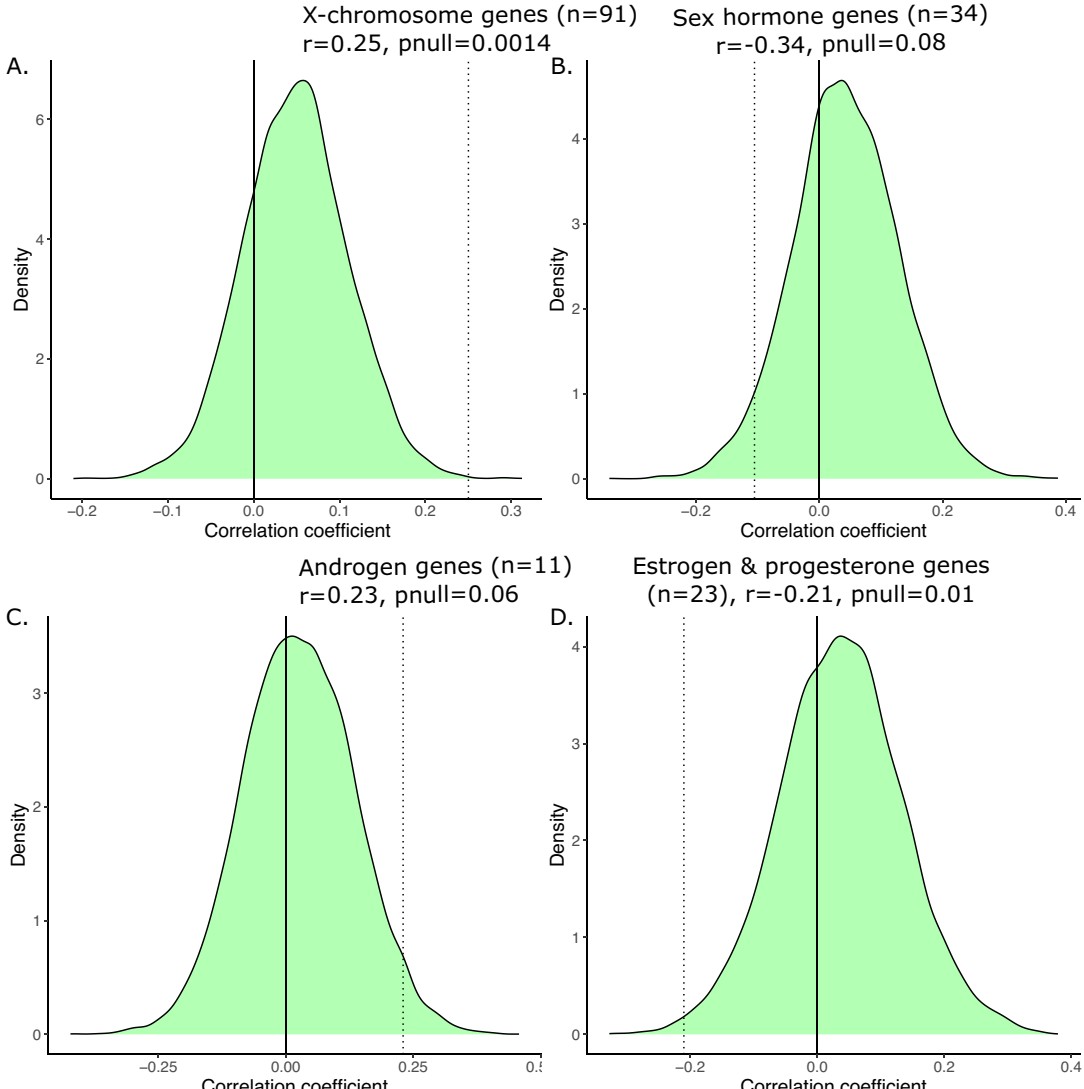

**Appendix 3—figure 2.** Generating a null distribution of correlation coefficients for the anatomical vs. transcriptional similarity. Correlations are significant relative to null distribution (shown in green) generated by randomly sampling subsets of 2835 homologous genes corresponding to the biologically informed subsets, recomputing the transcriptional similarity, and correlating that with the anatomical similarity 10,000 times. We have subsets of 91 genes in (**A**) corresponding to X-linked genes, 34 genes in (**B**) corresponding to sex hormone genes, 11 genes in (**C**) corresponding to androgen games, and 23 genes in (**D**) corresponding to estrogen and progesterone genes. For each, the observed correlation was compared to the null correlation to generate a p-value, displayed on the graph.

## Appendix 4

### Deformation-based morphometry for hypothalamus segmentation

T1-weighted structural MRI scans were converted to the MINC file format (https://www.mcgill.ca/bic/software/minc) and processed using the minc-bpipe-library preprocessing pipeline (https://github.com/CobraLab/minc-bpipe-library, copy archived at *Devenyi, 2024b*) which performs an iterative whole-brain bias field correction using N4ITK (*Avants et al., 2011*). Images were cropped to remove excess data around the head to improve subsequent image processing steps. A brain mask was computed using the BEaST patch-based segmentation technique (*Eskildsen et al., 2012*) which was used to generate skull-stripped brains for subsequent steps. Preprocessed, skull-stripped T1-weighted images were then registered using antsMulitvariateTempalteConstruction tools (https://github.com/CoBrALab/optimized_antsMultivariateTemplateConstruction). Briefly, this was used to affinely and nonlinearly register brains to create a consensus average. The initial target was the MNI ICBM NLIN 09 c model (1 mm isotropic resolution), but a subsequent study-specific average was created, and then upsampled to 0.5 mm isotropic. The registration procedure yielded deformation fields from which we can extract absolute Jacobian determinants that encode the contributions from the affine and nonlinear transforms between subject and average, as well as the relative Jacobians which have the affine (linear) component removed. Finally, Jacobians were smoothed with a 2 mm Full Width Half Maximum (FWHM) 3D Gaussian smoothing kernel.

# Appendix 5

**Appendix 5—table 1.** Information about the origin laboratory of wild-type (WT) mice was included in the study.

| Background strain | Study cohort key | Origin laboratory |
| --- | --- | --- |
| | A | University of Michigan; Dr. Diane Robinson |
| | B | KAIST; Dr. Eunjoon Kim |
| | C | University of Western Ontario; Dr. Nathalie Berube |
| | D | UT Southwestern; Dr. Genevieve Konopka |
| | E | Duke University; Dr. Christelle Golzio |
| | F | Duke University; Dr. Christelle Golzio |
| | G | Lost Angeles Children's Hospital; Dr. Pat Levitt |
| | H | Columbia University; Dr. Jeremy Veenstra-VenderWeele |
| | I | Scripps Research Institute; Dr. Gavin Rumbaugh |
| | J | McMaster University; Dr. Karun Singh |
| | K | McMaster University; Dr. Jane Foster |
| C57BL6J | L | University of Toronto Center for Phenogenomics |
| | M | The Hospital for Sick Children; Dr. Lauryl Nutter |
| | N | UC Davis - MIND Institute; Dr. Alex Nord |
| | O | The Hospital for Sick Children; Dr. Lauryl Nutter |
| | P | UCSD; Dr. Lilia Iakoucheva |
| | Q | UT Southwestern; Dr. Graig Powell |
| C57BL6N | R | UC Davis; Dr. Alexander Nord |

