## [Editor Report · eLife assessment]

In this **important** study, Guma and colleagues describe the use of structural neuroimaging to assess the cross-species convergence of sex differences in global and regional brain volumes in humans and mice. The goal of the work is to inform to what extent mouse studies of these aforementioned sex differences have relevance to humans. The authors suggest which aspects of brain anatomy (as measured by volume) are conserved or not, across species, which has theoretical and practical implications beyond a single sub-field. The evidence to support the findings is **solid**, it uses methods and data analysis that are appropriate and validated.

---

## [Referee Report · Joint Public Review]

Summary:

Guma and colleagues set out to compare to what extent differences in total and regional brain volumes, as measured by structural magnetic resonance imaging (MRI) are conserved or not, between humans and mice. The rationale for this work is to inform the best use of the mouse as a model system to carry out mechanistic studies of how sex differences arise in brain volumes, based on convergence to humans. This has practical implications for multiple fields in neuroscience. The authors find a modest convergence on these measures highlighting important areas for further mechanistic study.

Strengths:

The main strengths of the study lie in the use of a cross-species technology, i.e. structural MRI, using tools and methods that have been extensively validated.

Weaknesses:

Limitations of the study include, as acknowledged by the authors, the focus on a specific age range in mice and humans (which may not be congruent) and the lack of information regarding sex differences earlier or later in life. This has relevance with regard to the ages of onset for psychiatric and neurological disorders for example, which show apparent sex differences in prevalence. The paper also does provide data for an intermediate phylogenic level of analysis, such as data from primates. Lastly, these data do not provide any evidence as to the mechanisms underlying sex differences, when they arise, and to what extent they impact behavior.